

# Determination of the Atmospheric Volatility of Pesticides using Chemical Ionisation Mass Spectrometry

Olivia M.Jackson[1], Aristeidis.Voliotis[1,2], Thomas J.Bannan[1], Simon P.O'Meara[1,2], Gordon McFiggans[1], Dave Johnson[3] and Hugh.Coe[1]

[1]Centre for Atmospheric Sciences, Department of Earth and Environmental Science, School of Natural Sciences, University of Manchester, Manchester M13 9PL, UK
[2]National Centre for Atmospheric Science, Department of Earth and Environmental Science, University of Manchester, Manchester M13 9PL, UK
[3]Syngenta Jealotts Hill Research Station Jealotts Hill, Bracknell RG42 6EY, UK

*Correspondence to*: Hugh Coe (hugh.coe@manchester.ac.uk)

**Abstract.** Pesticides have been found to be transported through the atmosphere away from fields on application. A key indicator of a pesticide's likelihood to reside in the atmosphere is its vapour pressure. Within this study we evaluate a novel method, the Filter Inlet for Gases and AEROsols (FIGAERO) coupled with a chemical ionisation mass spectrometer using a set of calibration compounds, poly-ethylene glycols (PEGs). Two methods of compound delivery onto the filter have been

tested: atomisation and syringe deposition. Delivery results are consistent with previous studies, highlighting the lack of suitability of the syringe method. The successful calibration using the atomisation method was then used to determine the vapour pressure of 6 pesticides. This is the first-time particle phase pesticides have been measured with online mass spectrometry. The pesticides have then been compared to widely accepted standard literature values used in industry as well as values determined by a common environmental model also used in industry to give an indication of pesticide volatilities.

Results showed that measurements from the FIGAERO-CIMS were consistent with reported literature values for some compounds, others differed by up to 2 orders of magnitude. Determinations of Dicamba, MCPA and MCPP volatility using the FIGAERO-CIMS showed them to be semi-volatile in agreement with literature values to be within an order of magnitude. Mesostrione exhibited the largest difference in volatility with the FIGAERO-CIMS measuring a low volatility of $4.12 \times 10^{-8}$ Pa at 298K (compared to a literature value of $5.7 \times 10^{-6}$ Pa). The difference for 2,4-D of one order of magnitude perhaps can be

explained by the smaller particles deposited on the FIGAERO filter compared to the aerosolised PEG calibration particles, leading to evaporation at lower Tmax values and a lower measured vapour pressure. The atmospheric implications of the pesticide volatilities are also discussed. A pesticide's volatility is often a key indicator of the likelihood of the potential for short- or long-range transport occurring, thus determining a pesticide's fate in the atmosphere and potential for environmental pollution from transportation in the air.






## 1 Introduction

Pesticides are a group of compounds whose fate and behaviour in the environment and particular the atmosphere is less well studied and characterised in comparison to soil, surface water and ground water environments. (Socorro et al., 2016); The Food and Agriculture Organisation (FAO), a specialist body of United Nations, defines pesticides as follows: "Pesticides refer

to insecticides, mineral oils, herbicides, fungicides and bactericides, fungicides, plant growth regulators … other substance … for preventing, destroying or controlling any pest, including vectors of human or animal disease, unwanted species of plants or animals causing harm during … the production, processing, storage, transport or marketing of food, agricultural commodities … or animal feedstuffs …"

An important subset of pesticides is formulated plant protection products (PPPs) containing synthetic chemical active

substances which may be applied – e.g., sprayed as a water solution or emulsion – onto growing crops, at different stages of development of the plant. On application, there is direct exposure of environmental compartments – soil, water, air – in the vicinity of agricultural activity in, e.g., fields and orchards. Pesticides are stringently 'regulated' chemicals. In the EU, for example, Regulation (EC) No. 1107/2009 specifies requirements and conditions for approval, including that "A pesticides…. shall have no immediate or delayed effect on human health … it shall have no unacceptable effects on the environment …".

Concerning the latter, the Regulation further specifies that consideration should be made of "(i) … fate and distribution in the environment, particularly contamination of surface waters … groundwater, air and soil taking into account locations distant from its use following long-range environmental transportation; (ii) … impact on non-target species …; (iii) … impact on biodiversity and the ecosystem." These regulatory legal requirements must be met, and the safety and risk profile of a pesticide be deemed acceptable in order for a product to be sold on the market.

Many pesticide active substances are relatively large molecules, (molecular weights of 100s g mol$^{-1}$, which is typically large for most atmospheric compounds ) with low measured vapour pressures, and as such are commonly considered to be of negligible or low volatility , with a small selection deemed as semi-volatile (as defined by (Donahue et al., 2012a)). Following application, to the field or crop, the movement of pesticide material via the air is described as spray drift whilst post-application volatilisation may occur from soil or plant surfaces depending on field conditions and the pesticides physiochemical properties.

Typically, drifted pesticide will deposit with a few tens of metres from the location of spraying(Siebers et al., 2003), thus many environmental risk assessments consider impacts that occur within this distance. For a small number of PPPs airborne transport has been shown to be significant, with observations in the Arctic being measured (albeit with typical concentrations of a few pgm$^{-3}$)(Balmer et al., 2019). The majority of the latter are called 'legacy' pesticides – i.e. no longer in use, which have been categorised as a Persistent Organic Pollutant (POP) under the Stockholm Convention (Treaty Series No. 22 (2005)) and thus

are not permitted to be used but concentrations are still measurable in the Arctic due to previous use and their persistence in the environment (e.g. soil) and often historical overuse. Small concentration of currently used pesticides (CUPs) have been detected and continue to be monitored at the Arctic stations (Kallenborn et al., 2012). In terms of current EU regulatory context, pesticide active substances of non-negligible volatility, must not have the potential to undergo significant atmospheric



transport, defined in terms of an atmospheric half-life of less than two days – Here atmospheric half-life is defined with respect
to OH-radical-initiated, or O3-initiated, oxidation in the gas phase.

Pesticide active substances, by design, exhibit biological activity towards target pests. This also means that, intrinsically,
active substances may exhibit toxic and/or ecotoxic characteristics. Impacts or effects of the latter are considered via risk
assessments whereby estimated exposure levels are compared with quantitatively established levels, or thresholds, that indicate
absence of unacceptable effects. There can be a multitude of consequences to applying pesticides via spray, such as spray drift
to unintended crops, which can lead to damage and decrease in yield (Brochado et al., 2022). Health impacts have also been
reported in nearby populations due to spraying as well as an increase in reports of cancer observed in farm workers exposed
through dermal absorption or inhalation (Samanic et al., 2006; Lerro et al., 2020). The full magnitude of the exposure due to
spray drift is hard to estimate; one study estimated that exposure at work in the United States could be as high as 12-21 million
people per year (Lee et al., 2011). Therefore, a full assessment of these implications must be carried out and mitigations such
as the use of personal protective equipment (PPE) implemented in order to support safety cases for legal approvals of active
ingredients.

Where environmental exposure and risk assessment is well developed for surface water, groundwater, and soil environments
– and the organisms and ecosystem services therein – there has been relatively much less attention on the fate and behaviour
of pesticides in the atmospheric environment. There is growing interest and activity regarding the measurement of pesticides
in air. To properly interpret such data – including the potential for effects on humans, the environment, and agriculture due to
significant air-mediated pesticide transport – it is important to understand volatility to assess potential emission to air. Reliable
vapour pressure measurements are a key element of this, yet accurate measurements can be technically challenging for semi-
volatile chemical compounds or those of low volatility at environmentally relevant temperatures.

Volatility underpins the prediction of how likely a compound may reside in the air. It is then possible for a compound to
partition between the gas and atmospheric particulate phases after volatilisation, this in turn allows the estimation of aerosol
lifetimes. This, for example was recently studied in the case of the active ingredient difenoconazole, which has a measured
vapour pressure of ca. $3 \times 10^{-8}$ Pa, at 20°C (Socorro et al., 2016). This would suggest that if present in air, difenoconazole is
predominately expected to be in the particle phase making it effectively shielded from degradative gas phase oxidation. This
is turn allows the possibility of long-range transport to occur by attaching to particles that in principle could be subject to
atmospheric transport over significant distances.

Volatility can be characterised by the saturation vapour pressure ($V_p$) of a compound. Vapour pressure is defined as 'the
pressure exerted by a pure substance in a system at a given temperature containing only vapour and condensed phases of the
substance.' (IUPAC) It is an activated process thus highly dependent on temperature. Vapour pressure measurements can then
be used to determine the volatility of a compound (i.e., how likely it will be found in the gas phase). The measurement of a
pesticide active substance's vapour pressure is a regulatory data requirement, and the measurement can be used as a
physicochemical input parameter in regulatory environmental fate models.





Standardised validated methods have been developed for determining the vapour pressure of a chemical substances. In the EU, for example, regulatory requirements specify that that vapour pressure of the pesticide active substances should be determined in accordance with OECD method 104 (Oecd, 2006); this comprises 8 different measurement methods/techniques that may be

more or less suitable depending on the test compound, for example, the anticipated vapour pressure or the physical state of the sample. This includes a Knudsen effusion method (Booth et al., 2009) that has been previously been demonstrated to be applicable to atmospherically relevant compounds in the temperatures ranges from 15°C to 40°C. Another common method included in the OECD guideline is the gas saturation method (Widegren et al., 2015). Nevertheless, it is essential to recognize that there isn't a single universally applicable approach capable of encompassing all vapor pressures and temperatures, mainly

due to the challenges associated with validating vapor pressure measurements. In published pesticide regulatory literature, the specific method may not be stated and commonly measurements made at higher temperature are extrapolated to a more environmentally relevant reference temperature e.g., 20°C.

In the EU, as an example, currently "regulatory accepted" vapour pressure measurements for pesticides are derived from studies which were submitted by registration applicants and included as part of a dossier of data and assessments. Subsequently,

the information presented in the dossier is assessed and summarised in an official assessment report, the latter is then subject to a peer-review process which is co-ordinated by the European Food Safety Authority (EFSA). At the end of the peer-review process, EFSA publishes a "conclusion" document which contains a list of agreed risk-assessment "endpoints", including the vapour pressure of the active substance. This list does not contain study details, but this information is available from the official assessment report, which is also published and can be accessed via the OpenEFSA web portal. These endpoints, and

other information have been compiled and summarised as part of the University of Hertfordshire database Pesticide Properties Database (Lewis et al., 2016) providing an easy way to access this information.

Furthermore, there is a particular difficulty when measuring many atmospherically relevant compounds due to the low pressures required to be measured as well as being able to measure at temperatures close to those representatives of environmental conditions. Common techniques for this have been summarised (Bilde et al., 2015) using dicarboxylic acids to

compare across the techniques. Other measurement techniques have also been developed, and the vapour pressures of single compound aerosolised droplets have been measured using electrodynamic balance (EDB) and optical tweezer methods (Davies, 2019; Cai et al., 2015). Most current methods, measure volatility by considering single aerosols but measurements of bulk particles maybe important. One such approach is the Filter Inlet for Gases and AEROsol - Chemical Ionisation Mass Spectrometry (FIGAERO-CIMS) (Lopez-Hilfiker et al., 2014); in which the compound of interest is delivered onto a filter

which is then reverse flushed with nitrogen that is heated through a increasing temperature cycle to volatilise the compound. The volatilised substance is then detected by Chemical Ionisation Mass Spectrometry (CIMS).

Previous volatility literature has commonly concluded that different vapour pressure measurement techniques do not agree and often data spans several orders of magnitude for the same compound. This can be explained by the fact that each technique treats a substance differently in terms of pressure, temperature, and the phase state it is measured in making comparison

between different methods challenging and potentially erroneous. To begin mitigating this, a unified reference approach using



a range of polyethylene glycol (PEG) ($C_{2n}H_{4n+2}O_{n+1}$) compounds of different chain lengths was proposed and measured with a range of measurement techniques that were then evaluated. The PEG series was chosen since different polymer chain lengths have different vapour pressures from very volatile to largely involatile and includes the atmospherically relevant fraction (Krieger et al., 2018). Further work also has used the FIGAERO-CIMS  (Bannan et al., 2019) to characterise the calibration

of the PEG series with chain lengths from 1-8, which produced calibration curves in the atmospherically relevant vapour pressure range. In this work a compound of interest was syringed onto the FIGAERO filter. However,  subsequent work (Ylisirniö et al., 2021), showed that atomising the compound gave a more uniform dispersion of smaller droplets on the filter that are consequently evaporated more rapidly than larger syringed droplets, and so volatilisation occurs at lower temperatures than is observed using the syringe method leading to lower retrieved vapour pressures (Schobesberger et al., 2018).

Vapour pressure ($V_p$) has previously been related to, the saturation mass concentration in the gas phase, C*, which is calculated using Eq. 1**.** The C* of a compound represents the mass concentration at which 50% exists as a vapour and 50% will be in condensed equilibrium. This provides a framework for considering how components of a multi-component system, such as is present in the atmosphere, will be likely to partition between gas and particle phase and classifies compounds into volatility basis set (VBS) groups.


$$C^* = \frac{M_w V_p \gamma}{RT} \quad (1)$$

Here $M_w$ = molecular weight(gmol$^{-1}$), $\gamma$ = coefficient, R = universal gas constant, T = Temperature.

Pesticides commonly have C* values in the range of 0.3 - 300µg m$^{-3}$, typically classified as semi-volatile compounds (SVOC) (Donahue et al., 2012a; Donahue et al., 2006). However, due to the wide range of structures present in different pesticides some may be characterised as being of lower or higher volatility. A compound with C*< 0.3 µg m$^{-3}$ is considered a low

volatility organic compound (LVOC) whilst a compound with C* in the range 300 - 3× 10$^6$µg m$^{-3}$ is classed as a compound of Intermediate volatility compound (i.e. IVOC) (Donahue et al., 2012b). This provides a basis for estimating whether a compound is likely to be present predominately in the gas or particle phase under atmospheric conditions.

The aim of the present study was to demonstrate a standard procedure for calibrating the measurement of volatility using the FIGAERO-CIMS technique. This method is then applied to determine the vapour pressure of a set of common pesticides, the

measurements are then compared to regulatory literature and modelled values.

## 2    Methodology

### 2.1 Chemical Ionisation Mass Spectrometry with FIGAERO

A High-Resolution Time-of-Flight Chemical Ionisation Mass Spectrometer (HR-TOF-CIMS) was used with an iodide reagent ion source, a technique which has previously been used for both online and offline analysis of a range of oxidised organic

compounds (Yatavelli et al., 2012; Mohr et al., 2013; Voliotis et al., 2021; Bannan et al., 2019).



In order to study the particle phase, the FIGAERO inlet (Lopez-Hilfiker et al., 2014) was used in this study, which allows simultaneous gas and particle sampling using a dual inlet-system. The gas phase sampling inlet port allows ambient air directly into the CIMS Ion Molecule Region (IMR) where target molecules react with the iodide ions. The iodide reagent ions were produced by passing methyl iodide ($CH_3I$) and nitrogen ($N_2$) over a $^{210}Po$ radioactive source (Lee et al., 2014). Simultaneously,
the atmospheric particles are collected via the aerosol inlet onto a polytetrafluoroethylene (PTFE) filter. After a period of gas sampling and particle collection the filter is physically moved so that it is in line with the sampling line into the IMR. In this study only the particle phase is collected. The filter (and any particles on it) is backflushed with nitrogen and continuously heated at a rate of 8.75°C min$^{-1}$ to a programmed set temperature of 200°C. This is known as "the ramp", after which is held for 30 minutes using a backflush of heated $N_2$ gas, known as "the soak". The heat desorbs the particle phase compounds off
the filter into the vapour phase dependent on their volatility, the compounds are then transported via the nitrogen flow to be measured by the instrument. Analysis is performed using the Tofware data-analysis package (v3.2.3) in IGOR Pro in which the m/z of the analyte compound is selected and the ion count is plotted as a function of temperature to retrieve a thermogram. The temperature at which the peak signal for a given m/z occurs is labelled the $T_{max}$ for that ion peak, which can be related to the saturation vapour pressure provided that a suitable calibration peak is available to determine empirical constants. The
relationship between $T_{max}$ and saturated vapour pressure is given by eq. 2. In this study poly(ethylene) glycol (PEGs) were used as a set of calibration compounds (Krieger et al., 2018; Bannan et al., 2019).

$$\ln(V_{P,lit}) = a\,T_{max} + b \quad (2)$$

Two types of delivery of compounds onto the FIGAERO filter for introduction into the instrument were assessed: 1) syringe deposition 2) atomisation (Ylisirniö et al., 2021; Bannan et al., 2019). The syringe method consists of dissolving the analyte
into a volatile solvent (acetonitrile), then syringing 1-5µL of solution on the PFTE filter. The second method follows the approach by (Ylisirniö et al., 2021), which nebulises polydisperse particles of the analyte plus solvent (acetonitrile) using an atomiser, with the set up shown in fig. 1. The atomiser is connected to a 150L steel drum, through which a flow of 20 Lmin$^{-1}$ is passed, allowing evaporation of the solvent. The sampling time is dependent on the concentration of particles determined by the SMPS-CPC (Scanning Mobility Particle Sizer – Condensation Particle Counter) (Eq. 3). A SMPS-CPC (water-based),
sampling at 0.6 Lmin$^{-1}$ was used to monitor the particle mass concentration in the drum, to determine the total particle load. The SMPS was also used to monitor the sized distribution of the atomised particles. Previous work (Ylisirniö et al., 2021) recommended that through polydisperse particles may be used in the determination of vapour pressures using the FIGAERO, the particle size distribution of the calibration particles should be matched to that of the sampled particles so that the evaporation times are well matched to those of the calibration particles.

$$t = \frac{m_{filter}}{Q\,c_{SMPS}} \quad (3)$$

Here t = sampling time, $m_{filter}$ is the mass on the filter required (1µg), Q is the flow rate in m$^3$ min$^{-1}$ and $c_{SMPS}$ is the concentration of particles sampled by the SMPS in µgm$^{-3}$. Once sampling has taken place the FIGAERO temperature ramp can begin.



The aim of the present work is to better understand the fate of pesticide substances in the atmosphere; thus, the vapour pressures of several pesticide active substances were investigated. Firstly, a calibration method (Ylisirniö et al., 2021) using a PEG

mixture was established. The calibration was then used to determine the vapour pressure of the chosen pesticides.

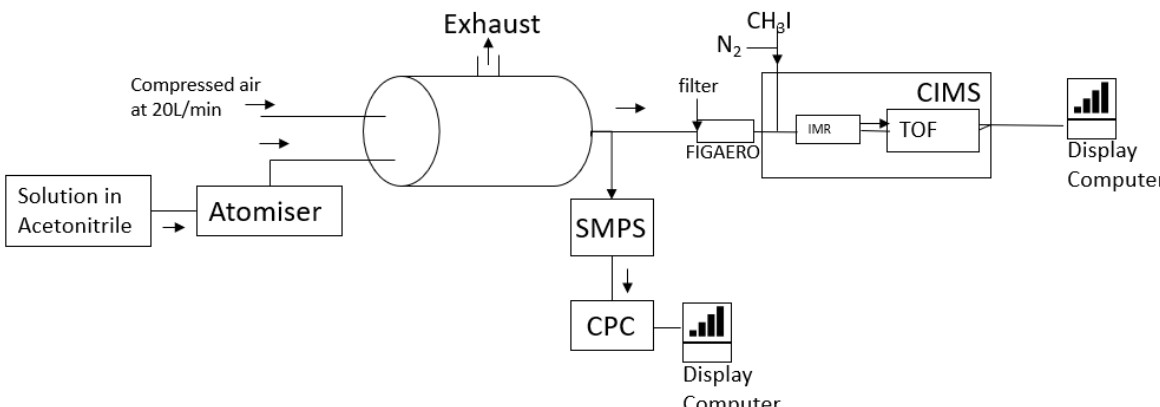

**Figure 1: Diagram of the atomisation set up used in these experiments using a 150L spacing drum before introduction into the CIMS instrument for detection and analysis.**

**2.2 Structure Activity Relationships (SAR)**

There are 100,000s of compounds present as atmospherically relevant aerosols therefore estimation methods are required since it is not possible to measure every compound. Several reviews have highlighted the complexity and varying degrees of success of these models (Barley and Mcfiggans, 2010; O'meara et al., 2014). Structure activity relationship models are commonly used as a first prediction and screening tool for a compound's physiochemical properties and thus environmental fate and behaviour (Dearden, 2003; Leistra, 2011). One such method used in this study is the Nannoolal et al method (Nannoolal et al., 2008)

where the group contribution ($dB$) and $T_r$ ($T_r = T/T_b$ where T is the modelling temperature and $T_b$ is the normal boiling point) are both calculated and input into eq. 4.

$$\log_{10}(V_{p,pred}) - (4.1012 + dB)(\frac{T_r - 1}{T_r - \frac{1}{8}}) \quad (4)$$

The Nannoolal model was chosen as training data for the model used a good proportion of aromatic compounds as well as considering a wide series of functional groups (O'meara et al., 2014; Barley and Mcfiggans, 2010).

The EPI suite (Estimation Programme Interface) created by the US Environmental Protection Agency (EPA) is commonly used in industry to predict a compound's physiochemical properties, as it includes several different models to predict different properties. Vapour pressure is calculated by inputting the compound structure through a SMILES string description, the software then searches a database to match known values (such as melting point) that are used to predict vapour pressure. The specific method used was the modified grain method (MGM), a modified version of the Grain-Watson method (GW)



(Vermeulen, 1991). MGM is commonly used for vapour pressures of solids (US EPA). In this work, the pesticides are dissolved in acetonitrile before nebulising and then the solvent evaporates after aerosol is generated assuming the remaining particles left to be in the solid phase. It is important to note that not all of the MGM methods are perfect for pesticide vapour pressure estimation due to the wide variety of different functional groups present within the different structures of pesticides.

## 2.3 Reference Compounds

The vapour pressures of PEG 1-8 ($C_{2n}H_{4n+2}O_{n+1}$) have been measured using multiple techniques including the FIGAERO CIMS. Previously liquid samples of one specific PEG of a defined chain length have either been measured individually or weighed out and added into a single solution containing a number of PEGs of different chain lengths, this is expensive and time consuming. The FIGAERO-CIMS method has the benefit of being able to measure thermograms (and thus vapour

pressure) of multiple compounds simultaneously at different m/z values therefore analyte solutions can contain multiple compounds, assuming that no interaction occurs between them.

In this work, PEG-400, an analytical standard reference material comprising a mixture of PEGs, was procured from Sigma Aldrich in order to eliminate the need for the individual solutions. PEG-400 contains a variety of chain length polymer units of poly(ethylene) glycol with a weighted average molecular weight ($M_w$) of 400gmol$^{-1}$. A single nebulisation of this solution

can produce a calibration curve containing polymer chain lengths beyond 8 and thus vapour pressures of LVOC compounds can be determined.

## 2.4 Selection of Study Pesticides

Initial screening of pesticides was carried out to ensure that there was a good response on the CIMS of the pesticide using the iodide regent. The iodide reagent ion has previously been found to be sensitive to organic compounds including those which

are highly oxygenated (Ye et al., 2021). Pesticides contain a variety of functional groups and inorganic species depending on their mode of activity (e.g., herbicidal, fungicidal, insecticidal), thus limiting and defining the scope of pesticides available to study.

Initial selection of pesticides was based on those previously measured in the literature using CIMS techniques. CIMS has been previously used in both the laboratory and in the field to measure gas-phase pesticides including Trifluralin, 2.4-D and MCPA

(See Table 1 for chemical structures) (Murschell et al., 2017; Murschell and Farmer, 2019, 2018). This is the first-time particle phase pesticides have been measured with online mass spectrometry.

Further selection was made based on the following factors that consider the potential environmental and health impacts associated with pesticides(Jepson et al., 2020). A recent study (Hulin et al., 2021) highlighted 90 substances of potential concern to French populations in the air by categorising pesticides by their use, emission potential, persistence in the air and

chronic toxicity. Previous observations of the pesticide in the environment were also used as selection criteria suggesting it has the possibility to be a persistent pollutant (Fuhrimann et al., 2020). For example in the case of Dicamba, a commonly used



herbicide where reports of off-field Dicamba drift have been speculated to lead to environmental contamination in the USA due to destruction of crops adjacent to the applied field (Galon et al., 2021) also stimulated inclusion in the present study. Conversely it is also important to determine the volatility of pesticides thought to be involatile (i.e., no chance of volatilisation

in the atmosphere) to ensure that there is no potential for atmospheric presence thus no further risk assessment in air is required.

**Table 1: List of pesticides used in this study. (Literature vapour pressure values taken from the University of Hertfordshire Pesticide Property Database (Lewis et al., 2016).**

| Pesticide | Substance Group | Molecular Weight/ gmol$^{-1}$ | Formula | CIMS detected Ion and M/z | Structure | Literature Vapour Pressure at 20 °C/ Pa | Estimated Volatility Class |
|---|---|---|---|---|---|---|---|
| **2,4-D (2,4-Dichlorophenoxy acetic acid)** | Alkychlorophenoxy acid | 220 | $C_8H_6Cl_2O_3$ | $C_8H_6Cl_2O_3I$ m/z 347 | | $9.00 \times 10^{-6}$ | SVOC |
| **Dicamba** | Benzoic acid | 220 | $C_8H_6Cl_2O_3$ | $C_8H_6Cl_2O_3I$ m/z 347 | | $1.67 \times 10^{-3}$ | SVOC |
| **MCPA (2-methyl-4-chlorophenoxyacetic acid)** | Aryloxyalkanoic acid | 200 | $C_9H_9ClO_3$ | $C_9H_9ClO_3I$ m/z 327 | | $4.00 \times 10^{-4}$ | SVOC |
| **MCPP (Mecoprop-P)** | Aryloxyalkanoic acid | 214 | $C_{10}H_{11}ClO_3$ | $C_{10}H_{11}ClO_3I$ m/z 341 | | $2.3 \times 10^{-4}$ | SVOC |
| **Mesotrione** | Triketone | 339 | $C_{14}H_{13}NO_7S$ | $C_{14}H_{13}NO_7SI$ m/z 466 | | $<5.7 \times 10^{-6}$ | SVOC |

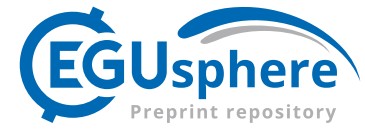

| Trifluralin | Dinitroaniline | 335 | $C_{13}H_{16}F_3N_3O_4$ | $C_{13}H_{16}F_3N_3O_4I$ m/z 462 | | 9.5x10⁻³ | IVOC |
|---|---|---|---|---|---|---|---|

**Table 1** shows the 6 pesticides selected in this study, all of which are pre-emergence herbicides, which specifically target

weeds present in the field post-harvest or pre-planting. All 6 pesticides are still commonly used worldwide. Trifluralin is banned in the EU (but still widely used in USA and Canada) due to concerns over its toxicity to aquatic organisms (Efsa, 2009; Waldbi, 1998) and previous reports have highlighted concerns relating to volatility- it's vapour pressure of $9.5\times10^{-3}$Pa classifies it as IVOC. On the other hand, Mesotrione is much less volatile with a literature vapour pressure of $<5.7\times10^{-6}$Pa indicating very low likelihood of appreciable atmospheric transport through volatilisation.

2,4-D, Dicamba, MCPA and MCPP all have similar structures. They contain chlorinated phenyl groups attached to an acid group and thus would be expected to have similar ranges of volatility with the different structures leading to relatively small differences in their physiochemical properties. Current vapour pressure predictions class these pesticides as SVOC. However, it must be noted that in a PPP 2,4-D may be present in different forms, the acid, ester, or salt in which the ester and salt form are derivatives from the acid form which is the active ingredient. For the salt form, it is a strong acid (pKa = 3.4 (Lewis et al.,

2016)) and thus highly water soluble and will be present as the acid form in hydrated environments (e.g., when the PPP is mixed with water prior to application, or in soil). Here, the vapour pressure is not expected to be impacted. 2,4-D has a reported vapour pressure of $9.00\times10^{-6}$ at 20°C(Lewis et al., 2016), which is the same as the acid form. However, for 2,4-D in the ester form (2,4-D- ethylhexyl ester) the vapour pressure has been reported as $4.8\times10^{-4}$ Pa at 25°C. In this study the acid form was chosen due to being the actual active species. This is because on application the ester form will cleave at the ester linkage.

Hence the pesticidal active chemical species is the anion of the acid.

## 3   Results

### 3.1 Calibration of the FIGAERO-CIMS Vapour Pressures using Poly(ethylene) glycols and Comparisons to Previous Studies

The FIGAERO-CIMS vapour pressure calibration was determined using Poly(ethylene) glycol - 400 in acetonitrile by first determining the $T_{max}$ of the PEG compounds of various chain lengths (n) from 1-16. Calibrations were performed with both the syringe and atomisation methods. The raw thermograms from the calibrations (with a normalised ion count), are shown in fig. 2. It can be seen that the $T_{max}$ increases with increasing PEG chain lengths. Since the volatility of PEG of a given chain



length is well known, a calibration curve can be constructed. Figure 2 also shows the difference in resolution achieved by the
syringe and atomisation methods. The atomisation method is unable to resolve PEG lower than C-5 as lower weight PEGs are
too volatile and may evaporate from the aerosolised acetonitrile solution in the drum, are thus not deposited on the filter as
particles and a peak in the thermogram is not observed over the measured temperature range. This is not the case for the syringe
method in which the solution is injected in bulk.  It can also be seen that the thermograms in the syringe method exhibit longer
tails after the peak due to the less uniform evaporation off the filter meaning some of the compound is evaporating off the filter
at much higher temperatures than the $T_{max}$ (Schobesberger et al., 2018). Conversely the thermograms in the atomisation data
are more uniform thus a more representative $T_{max}$ can be extracted.

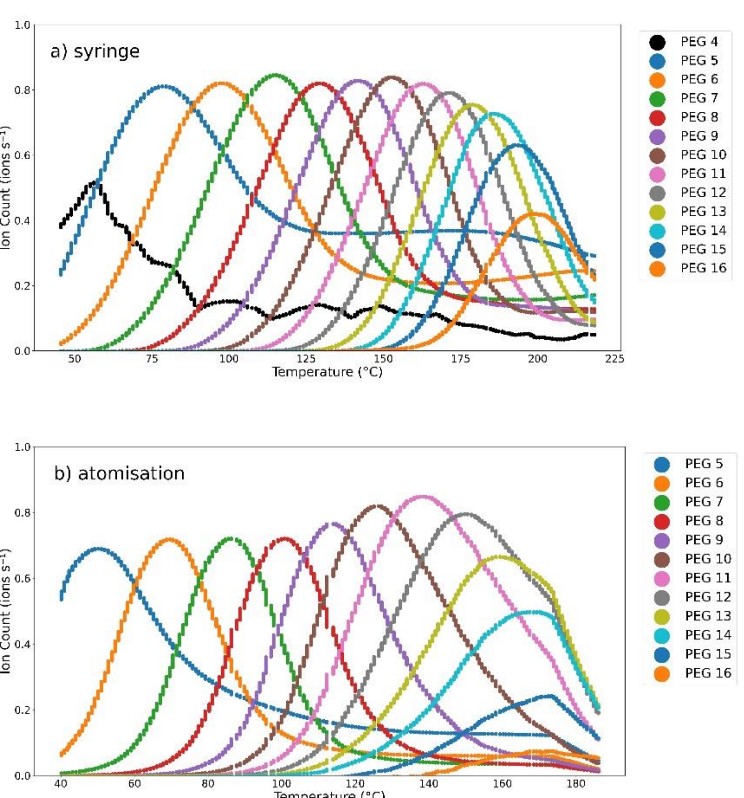


**Figure 2: Thermograms of the introduction of the PEG-400 calibration solution with normalised ion counts where a) syringe
method, b) atomisation method.**

Figure 3 shows how the $T_{max}$ values vary as a function of PEG chain length. The $T_{max}$ of the atomisation values are lower than
that of the syringe method as previously shown by (Ylisirniö et al., 2021).The $T_{max}$ values are compared with the literature
vapour pressures of the PEG series (Krieger et al., 2018) and are shown in fig. 3b.





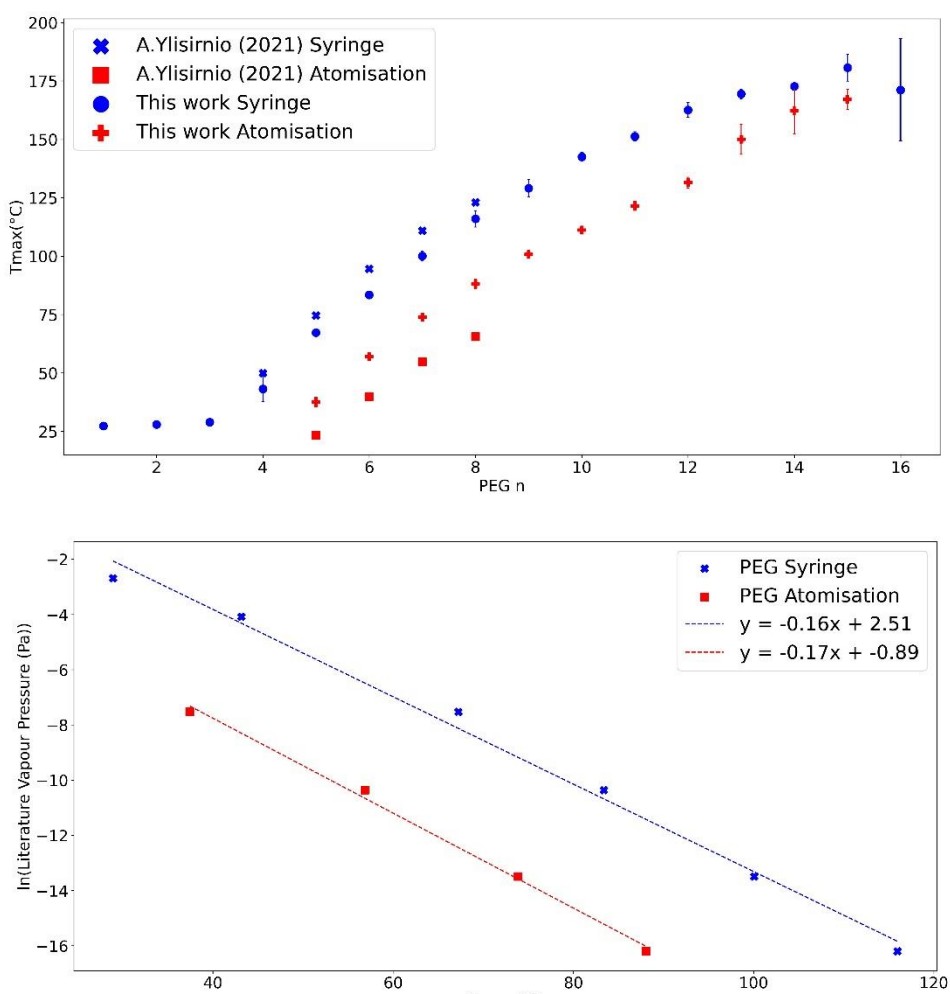


**Figure 3: a) T$_{max}$ values of PEG of varying chain length from this work and also determined by A. Ylinsirniö et. al. 2021. Syringe concentrations were 0.1 gL$^{-1}$ of PEG-mixture whilst atomisation measurements used sampling of an approximate concentration of 0.5gL$^{-1}$. All values from both sets of data came from an average of 3 measurements. b) the resulting calibration curves using both methods in this work using literature values from U. Krieger et al.,2018.**


The results are in line and expand those of (Ylisirniö et al., 2021), the T$_{max}$ values of the same PEGs derived from the syringe delivery method are substantially higher compared to when the aerosolised delivery method is used. The long evaporation rates observed when using the syringe method mean that the relationship between T$_{max}$ and reference volatility is strongly dependent on the size of the syringed droplets and will result in measurement bias unless the conditions of the test substance



match those of the calibration compounds very closely. Given the approach is subject to operational uncertainty we follow

the recommendation of (Ylisirniö et al., 2021) and use the aerosolization method to determine pesticide vapour pressures.

It can be seen in fig. 3a, there is a slight difference in $T_{max}$ values for PEG 5-8 between the previously reported observations

and those in this work that is most noticeable in the atomisation delivery method. This is explained by the difference in particle

sizes used in the nebulisation of the calibration particles in this work and the previous work. (Ylisirniö et al., 2021) explored

evaporation times using different sizes of monodispersed particles and showed a small but consistent repeatable effect of

particle size on the retrieved $T_{max}$ values for a given PEG compound. It was recommended that since the effect was modest,

polydisperse particles could be used for both calibrating the FIGAERO for vapour pressure and for determining vapour

pressure of the sampled aerosol components. However, Ylisirniö et al., (2021) recommend that the size distribution of the

calibration particles is matched to that of the samples particles whose vapour pressures are to be determined so that the

evaporation rates of sample material are similar to that of the calibration material with a similar volatility. In these experiments,

the particle distributions were measured with an SMPS to monitor for any differences (fig.4).

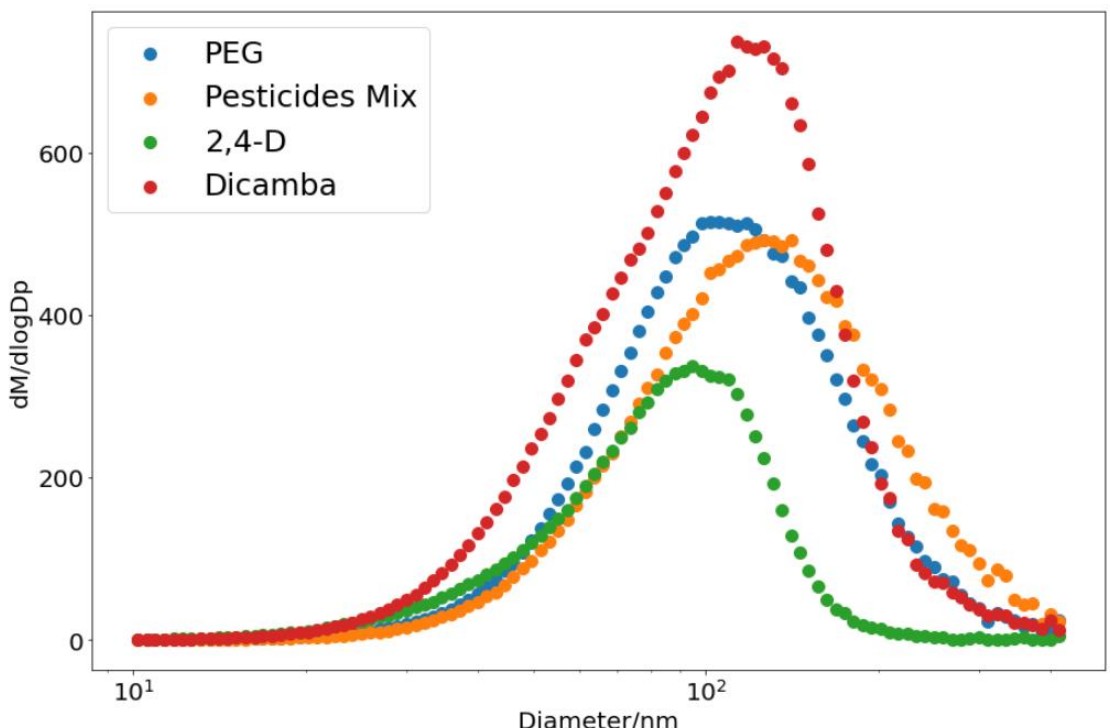

**Figure 4: Example set of size distributions for all the different solutions measured taken from the SMPS-CPC measurements during
the collection onto the FIGAERO filter. Where Pesticide mix includes MCPP, MCPA, Mesotrione and Trifluralin dissolved in**
**acetonitrile. 2,4-D and Dicamba are measured  separately as they appear the same m/z in the mass spectrum.**



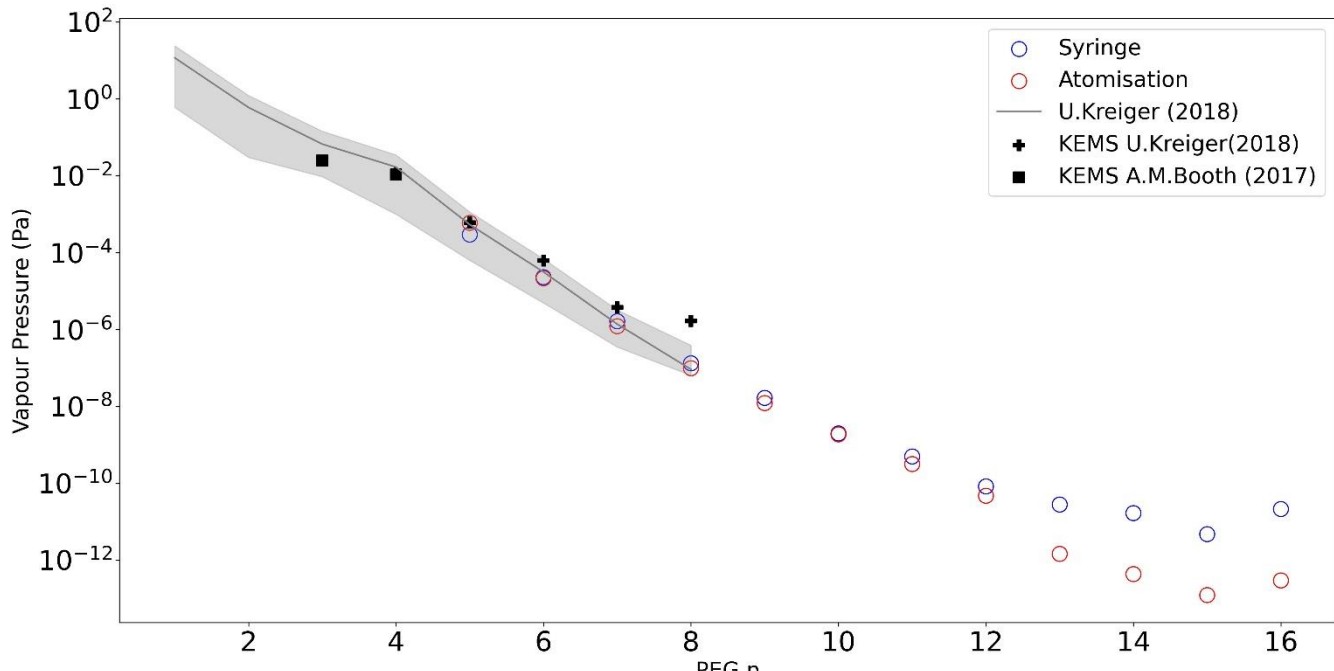

**Figure 5: Vapour pressures of PEG 2- 16 by PEG number from this work, U. Krieger et al., 2018 and A.M. Booth et al., 2017.**

The PEG vapour pressures from this work are shown in fig. 5 alongside the (Krieger et al., 2018) recommendation used to
derive the FIGAERO-CIMS calibrations together with the estimate of uncertainty based on the range of different methods
used over the range PEG 1-8. The FIGAERO-CIMS data in the overlap region (C5-8) are within the uncertainties of the Krieger
*et al.,* 2018 data, which is to be expected as this is the basis for the FIGAERO-CIMS calibration. In addition a comparison
with data from Knudsen Effusion Mass Spectrometer measurements (Booth et al., 2017) and Krieger *et al.*, 2018 are also
shown to be consistent with our data though some deviation in the lower vapour pressures is seen, which is likely due to the
KEMS being less sensitive to lower vapour pressure compounds. The experimental values from the present study significantly
extend the range of PEG vapour pressure measurements in the literature to much lower values, with a logarithmic slope
extending to vapour pressure measurements as low as $10^{-10}$Pa (PEG 12). Beyond, PEG-12 the variability in the data is likely
because these compounds are not fully evaporated from the filter at 437K (200°C). On the other hand, chain lengths lower
than PEG-4 are likely to be very predominately in the vapour phase, which is consistent with the observations that they fully
evaporate off the filter prior to the beginning of its heating cycle.



## 3.2 Determination of the Volatility of a Range of Pesticides by FIGAERO-CIMS and Comparison with Other Approaches

**Figure 6: Thermograms of the pesticides taken from the heating of the filters of the FIGAERO-CIMS measurements. The filters were collected using the atomisation delivery method and the thermograms are normalised to 1.**






**Figure 7: Tmax values of the pesticides taken from the heating of the filters of the FIGAERO-CIMS measurements. The filters were**
**collected using the atomisation delivery method.**

The PEG calibration data presented enabled the determination of the vapour pressure of a set of atmospherically relevant pesticide compounds. In the experiments Mesotrione, MCPA, MCPP and Trifluralin were measured in one solution dissolved in acetonitrile. However, 2,4-D and Dicamba both have the same molecular formulae (thus the same parent ion m/z value),

making it impossible to separate the thermograms when used in the same solution and therefore were measured in separate solutions. Figure 7 shows the $T_{max}$ values determined for each of the 6 pesticides, as the atomisation method has been demonstrated to be the most appropriate, only data from the atomisation delivery is shown. Here it can be seen that the highest $T_{max}$ was for the least volatile pesticide, Mesotrione and the lowest $T_{max}$ was for the most volatile pesticide, Trifluralin.



As reported previously (Ylisirniö et al., 2021) and discussed above, it is important to match the size distribution of the
calibration particles and those used in the vapour pressure experiments to minimise the influence of sample condition
differences inducing changes in evaporation times and hence $T_{max}$ values. Figure 4 shows the particle size distribution for each
of the solutions used in this work. Since evaporation times are faster for smaller particles deposited on the filter, $T_{max}$ values
will increase as particle size increase for particles of the same volatility.  Hence if the nebulised pesticide particle distributions
do not match that of the calibration size distribution, then biases in the vapour pressure determination are to be expected. This
may well be the case for 2,4-D from fig. 4.

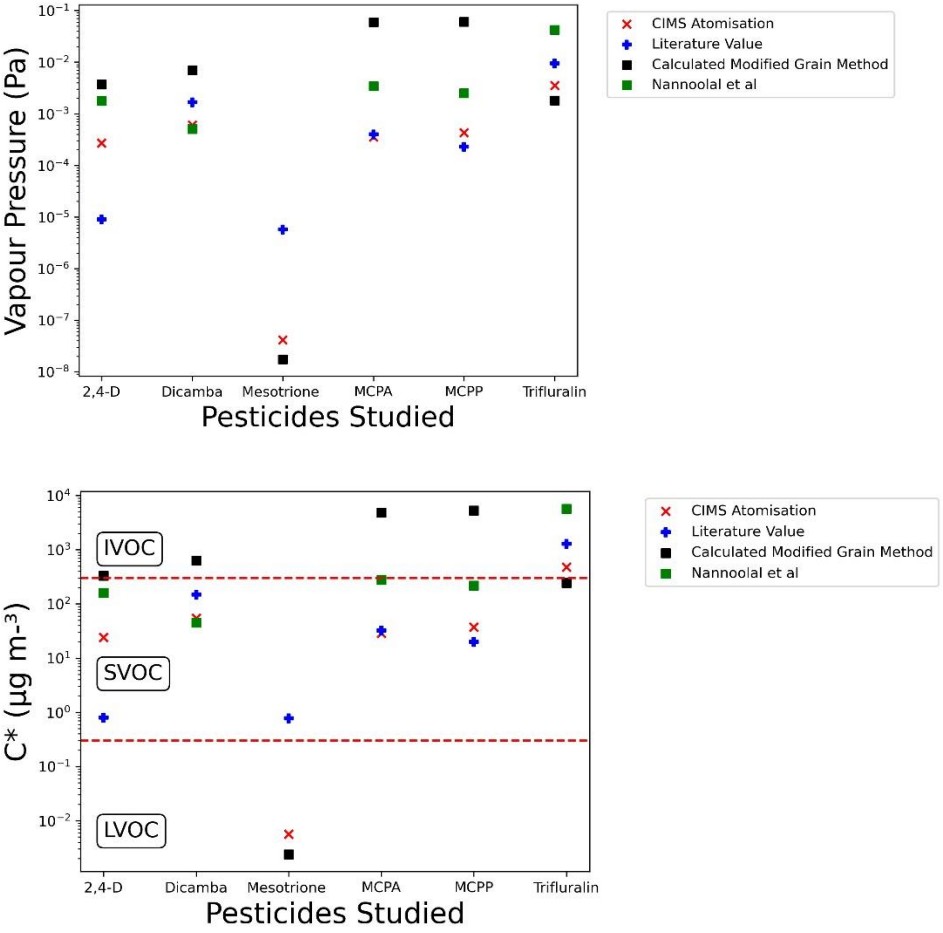

Figure 8: Measurement derived vapour pressure and C* values of pesticides determined from $T_{max}$ data using the calibrations
reported in section 3 and compared with accepted literature values, the Nannoolal et al method and the Modified Grain Method. .
The red dotted lines represent the boundaries between the indicative volatality basis (VBS) categories defined by (Donahue et al.,
2012b).



The $T_{max}$ values in Figure 7, were then used to derive vapour pressure values for each compound using calibration constants determined using the aerosolisation approach (fig. 3b). These vapour pressure values were then converted into C* values and

shown in fig. 8b, in order to make predictions of equilibrium gas-particle phase partitioning behaviour. Figure 8 also compares the data obtained by the FIGAERO-CIMS with literature values. These include currently accepted  regulatory endpoint values (e.g., as provided in the EU dossier for each pesticide); the Modified Grain Method (MGM), a model that is commonly used to provide initial predictions of a pesticide volatility based on SAR and the Nannoolol model another newer SAR model.

The FIGAERO-CIMS measurements show Trifluralin to have the highest vapour pressure and Mesostrione the lowest. 2,4-D

acid, Dicamba, MCPA and MCPP which all have similar structures all have similar measured vapour pressures (all within a magnitude of $1\times10^{-4}$ Pa). All 4 of these pesticides remain characterised as a SVOC (as seen in fig. 8b). Broadly these results are consistent with the MGM predictions. However, the MGM SAR tool predicted these herbicide molecules to be more volatile than observed with the CIMS experiments and the stated literature values by between 1 and 2 orders of magnitude. This confirms the importance of measured vapour pressure data.

Mesotrione exhibits the greatest discrepancy between the measured values and the comparison data. A vapour pressure value of $5.7\times10^{-6}$ Pa is stated in the University of Hertfordshire Pesticide Properties Database for Mesostrione; this is two orders of magnitudes higher than the value of $4.12\times10^{-8}$Pa determined with the FIGAERO-CIMS method. On further investigation of the regulatory literature value (i.e., looking into the renewals assessment report that sits behind the EFSA publishes list of endpoints (Efsa, 2016) and the EFSA conclusion itself). The vapour pressure stated in the literature (at 293K ) is based on a

value measured at 100.7°C (373K) and hence that the actual vapour pressure at 293K will be less than $5.7\times10^{-6}$ Pa at 293K (Efsa, 2016). This can be seen in Fig. 9, which shows the variation in vapour pressure with temperature in the form of a Clausius-Clapeyron relationship. In other words, the value of $5.7\times10^{-6}$Pa is considered to be an upper limit value. The observations at ambient temperatures presented in this work show Mesotrione to be considerably less volatile than may be expected based on the upper limit value. It can be seen that the reason for the difference in the EFSA document (Efsa, 2016)

is that the method used observed some thermal decomposition of Mesotrione (the melting point temperature is stated to be ~165°C/438K "with some decomposition on melting). This does not impact the FIGAERO-CIMS measurements which relies on thermal decomposition.

No value from the Nannoolol model is reported for Mesotrione as the sulphonyl group is not included in the model. Meaning any estimate using Nannoolol would not include the impact on vapour pressure of the sulphonyl (Nannoolal et al., 2008). On

the other hand, MGM SAR predictions gave a value of $1.75\times10^{-8}$ Pa. This was calculated using  theoretical melting point estimations and thus represents a predicted vapour pressure at 20°C, and so is more likely to be a more representative value than the current literature value that is widely used based on a conservative 20°C vapour pressure endpoint assumption since



it was not measured at 20°C. Data from the regulatory study, the present FIGAERO-CIMS study, and the MGM-SAR estimate are represented in the Clausius-Clapeyron plot (fig.9).

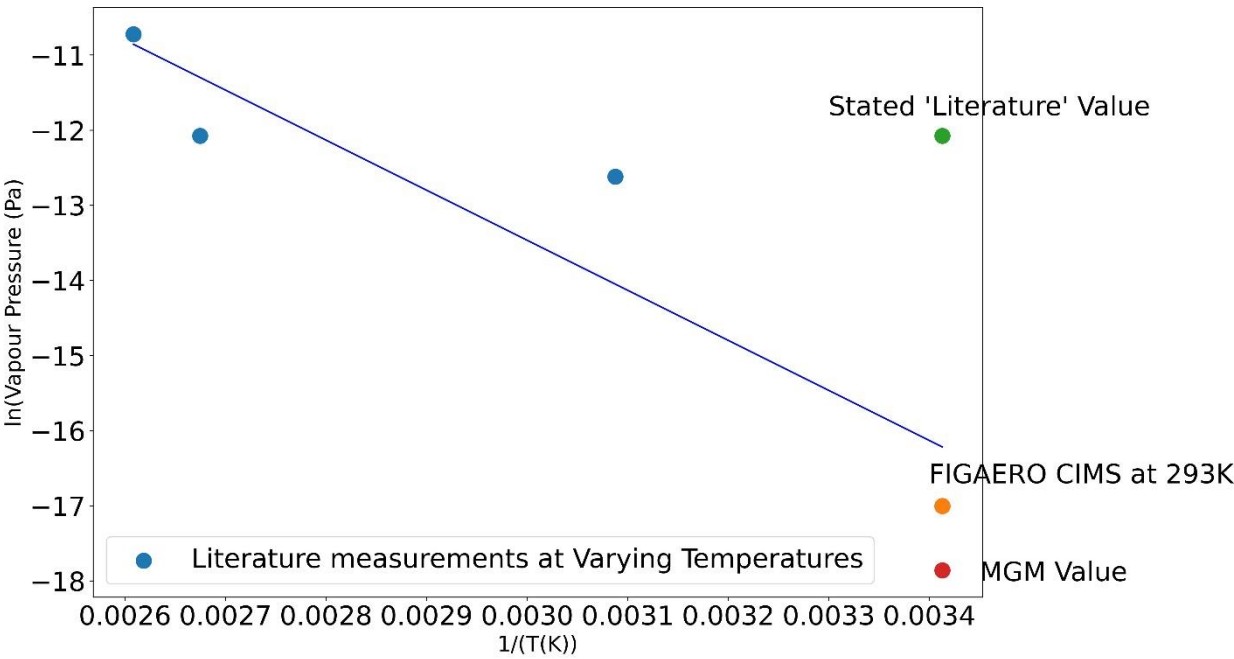

**Figure 9: Clausius Clapeyron plot of Mesotrione literature measurements compared with the CIMS atomisation measurement and the Modified Grain Method (MGM). Points in blue come from the EFSA endpoints (Efsa, 2009).**

The Clausius Clapeyron relationship for Mesotrione (Fig. 9) shows that the CIMS atomisation value obtained would fit well with the MGM predictions and strongly supports the value of $4.12 \times 10^{-8}$ Pa at 293K being more representative than the current literature value. Despite the current regulatory risk assessment endpoint vapour pressure differing significantly from the value from the present study, it does still indicate that volatilisation of Mesotrione is not expected to be significant in the context of the environmental fate and behaviour of this molecule.

Trifluralin is the most volatile of all the pesticide active substances investigated in the present study. A value of 27.7°C was measured as the $T_{max}$ which is close to the instrument limit due to the FIGAERO heating ramp starting at room temperature (approximately 25°C, meaning a $T_{max}$ value of below 25°C would not be measurable. This results in Trifluralin having a low sensitivity in the particle phase measurements of the CIMS due to potential volatilisation off the filter into the gas phase prior to heating. Despite these limitations a vapour pressure of $3.52 \times 10^{-3}$ Pa was measured, which is within one order of magnitude of the literature value ($9.5 \times 10^{-3}$ Pa). Conversely, the MGM-SAR gives a prediction of $1.8 \times 10^{-3}$ Pa, this is a closer prediction to the literature and measured values compared to the other pesticides, possibly suggesting the model shows better agreement with higher volatility compounds, whilst the Nannoolol model predicts a higher volatility, potentially due to the contribution from multiple functional groups.



## 4    Discussion

### 4.1 Assessment of the Calibration Method

A series of PEGs of different chain lengths and hence different volatilities were introduced onto the FIGAERO-CIMS filter.
From this a $T_{max}$ was determined for each PEG length chain from the temperature at which the maximum mass spectrum signal
was detected at the relevant m/z. The regression of $T_{max}$ with the literature values of PEG vapour pressure provided a calibration
for determining the vapour pressure of six representative pesticides. In this manner, the results of previous work were well
verified (Bannan et al., 2019; Ylisirniö et al., 2021) demonstrating the effectiveness of using atomisation as the delivery
mechanism for both the calibration and the measured compound onto the FIGAERO filter.

Biases occur when using the syringe method since the applied droplet sizes are much larger and take longer to evaporate, this
is within time scales that similar to or longer than the temperature ramp rate of the $N_2$ passing through the FIGAERO filter.
This potentially introduces biases since the differences in drop size from one syringing to another lead to shifts in Tmax
between different experiments for compounds of the same volatility.  This was previously demonstrated by (Ylisirniö et al.,
2021), where SEM images (Scanning electron microscopy) of the filters delivered using the two  delivery methods revealed
the syringe method deposited analyte in clumps whilst atomisation onto the filter gave a more even spread of particles on the
filter. The even distribution of particles on the filter evaporates more quickly and more evenly when heated than the clumps
formed when syringing due to a higher surface-to-volume ratio (Schobesberger et al., 2018).

One key point to highlight was that the concentrations on the filter were also replicated ($0.1gL^{-1}$ for the syringe and $0.5gL^{-1\ for}$
the atomisation method). This was done as previous studies (Bannan et al., 2019) found that a change in concentration
deposited on the filter produced a notably different volatility due to the evaporation rate differences of the analyte off the filter.
The use of PEGs as reference compounds for vapor pressure assessments was initially reported in the work of (Krieger et al.,
2018). Krieger and other prior investigations relied on singular PEG 3-8 solutions or combining the individual PEGs into one
solution and then dissolving them in a solvent. Previously, each PEG-n had to be individually weighed from separate containers
and added to the solvent. The PEG-400 mixture used in this work offers a wider spectrum of polymer chain lengths that
provides a robust multi-point calibration mixture in a single solution. Additional benefits include substantial cost savings and
reduction in wastage, increasing efficiency. Mass spectrometry detected PEG-1 to PEG-16+ within our analysis. However,
vapour pressures from the 1-5 and 13-16 ranges were not presented due to reduced repeatability. The shorter chain-length
inconsistencies (in PEGs 1-5) are likely attributed to highly volatile substances beginning to evaporate at ambient temperature
prior to measurement, as the 'ramp' phase initiates at approximately 25°C and thus would not be measured if the compound
had already evaporated into the gas phase prior to the ramp. For compounds with lower volatility (beyond PEG-12), the
temperatures reached during the FIGAERO ramp phase may not be sufficient to uniformly volatilize these compounds from
the filter. It is important to note that these limitations are limited to volatility ranges that mean the phase of the compounds
will be dominated one or other of the gas or particle phases and is unlikely to demonstrate significant partitioning behaviour.



Figure 5 additionally illustrates a logarithmic, decrease in the vapour pressure of the PEG series, indicating the robustness of
the FIGAERO-CIMS method in the pressure range $10^{-3}$ – $10^{-10}$ Pa (i.e., for PEG-5-12) without extrapolation. Lower volatility
compounds are unlikely to be atmospherically relevant if applied to the surface in the liquid or solid phase. This is seen for the
pesticide Trifluralin which exists predominately in the gas phase and whose vapour pressure was calculated to be $1\times10^{-2}$ and
yet still allows a good estimate to be determined by the FIGAERO-CIMS.

## 4.2 Atmospheric Implications of Pesticide Measurements


**Table 2: Calculated C\* values using the atomisation delivery method.**

| Pesticide | C* /µgm$^{-3}$ | Log(C*) / µgm$^{-3}$ |
|---|---|---|
| 2,4-D | 24.27 | 1.4 |
| Dicamba | 54.42 | 1.7 |
| Mesotrione | 0.0056 | -2.3 |
| MCPA | 28.82 | 1.5 |
| MCPP | 37.43 | 1.6 |
| Trifluralin | 476.4 | 2.7 |

Each pesticide's volatility, determined using the FIGAERO-CIMS was compared to the EU regulatory literature value and
two models based on structure activity relationships (MGM and Nannoolol) who's accuracies have previously been compared
(Barley and Mcfiggans, 2010). Both models clearly overestimate the vapour pressures of 2,4-D, MCPA and MCPP, predicting
more volatile vapour pressures than all of the literature values and CIMS measurements for these pesticides. This suggests that
the chlorinated benzene or carboxylic acid function groups may have been poorly represented in the training data set used to
develop the model. Conversely, Trifluralin estimations were predicted to be less volatile in the MGM and more volatile in the
Nannoolol in comparison to the other models, suggesting that the two models deal with the three functional groups (NO$_2$, NR$_2$
and CF$_3$) differently.  We expect the Nannoolol model to have more reliable estimations as the 'fishtine factor' can only take
6 values in the MGM but over 130 in the Nannoolol making the MGM limited when multifunctional groups are present (Barley
and Mcfiggans, 2010).

Some of the literature values, sourced from the EU regulatory literature exhibited disparities when compared to the FIGAERO-
CIMS values. These regulatory literature values were extracted primarily from their risk assessment or risk assessment review
of the compound. They are frequently employed in predictive models for estimating a compound's volatility. The regulatory
document (e.g.: (Efsa, 2016, 2009)) mentions the use of the OECD method 104, as previously described. However, it's
important to note that the OECD text guideline 104 encompasses several distinct methods for measuring vapor pressure.





Unfortunately, the reports do not specify which particular method was utilized. Consequently, it cannot be assumed that the pesticide literature values can be completely reliably compared due to the substantial variations in these methods.

Mesostrione exhibited the lowest volatility among the tested pesticides, yielding a C* value of $5 \times 10^{-3}$ µg/m³ (from a vapour pressure of $4.12 \times 10^{-8}$ Pa). This value stands nearly two orders of magnitude apart from the upper limit of the reference value of $<5.7 \times 10^{-6}$ Pa. This would shift its classification from a SVOC to a LVOC. The significant difference in vapor pressures can be attributed to this cautious regulatory value (taken at 100.7°C) in which from a regulatory assessment allows an absolute worst case scenario volatilisation prediction. This work then confirms Mesotrione is not prone to volatilisation and confirms

that larger portion of the sprayed compound will likely remain on plants and soil and undergo other pathways of degradation. For example, it is known that as a pesticide moves through the soil environment in particular it is assumed that it will not volatilise. Consequently, it becomes imperative to prioritize measurements related to the movement of Mesotrione in soil and water (Carles et al., 2017). From a regulatory standpoint, it's crucial to consider all potential pathways of environmental movement. It is also essential to clarify that this conclusion doesn't imply that Mesostrione is entirely free of environmental

risks. Rather, it suggests that it is very unlikely volatilisation is a major degradation mechanism. It may be however, undergo atmospheric transport if present on resuspended soil particles (Socorro et al., 2016).

Among the tested pesticides, Trifluralin was found to be the most volatile. The FIGAERO-CIMS analysis revealed only a one-order-of-magnitude difference from the literature value, supporting its IVOC categorisation (in regards to the scheme (Donahue et al., 2012a)). This classification suggests that Trifluralin is likely to quickly volatilise into the gas phase upon

application however is expected to quickly undergo chemical transformation, as reflected by the atmospheric life time of trifluralin being 0.45 days (with respect to OH-initiated photooxidation) (Efsa, 2009). Despite this a study in Arctic monitoring stations found low levels of Trifluralin in arctic air(Balmer et al., 2019) and has now been predicted that small amount of Trifluralin may stick to aerosol particles and transported significant distances. This potential for long-transport resulted in a major contribution to the removal of Trifluralin's approved status for use in the EU (Efsa, 2009); as well as it being consistently

ranked high in assessments of inhalation exposure risks, including potential health issues such as cancer (Sugeng et al., 2013; Coleman et al., 2020) and deemed extremely toxic to aquatic systems (Efsa, 2009). This resulted in the Trifluralin being considered to be added as a POP (Persistent Organic Pollutant) (EC 2007). Despite these concerns Trifluralin is still used in the US, Canada, and Australia it is consequently still important to understand the overall atmospheric fate of Trifluralin once it is present in the atmosphere.

The measurements of the vapour pressure of 2,4-D acid ($2.72 \times 10^{-4}$ Pa) resulted in a factor of 30 compared to the regulatory literature value $9.00 \times 10^{-6}$ Pa. Comparatively, there is an even larger difference of nearly 3 orders of magnitude ($3.72 \times 10^{-3}$ Pa) in the predicted vapour pressure by the MGM calculated method compared to the current regulatory literature value. The FIGAERO-CIMS measurements and MGM modelling would suggest 2,4-D is more volatile than previously stated. However, we note that the particle size distribution of 2,4-D particles deposited to the FIGAERO filter (fig. 4) was smaller than that of

the PEG calibration particle size distribution and had a significant tail at low diameters. Other pesticides size distributions had similar shaped distributions and peaks occurred at similar diameters compared to the calibration particles. Differences of this



magnitude have been previously reported to impact the $T_{max}$ values and thus alter the vapour pressure by up to half an order of magnitude as a result of changing the evaporation times on the filter (Ylisirniö et al., 2021). It also must not be forgotten that in the field an applied formulation 2,4-D is commonly prepared as either an acid, salt or in an ester form, with the acid and salt

dissociating into the anion form in hydrated environments. In Comparison to other methods, the FIGAERO-CIMS method is selective to the acid form as the thermogram at the m/z of 2,4-D is used to determine volatility and therefore it is certain the acid active substance is what is measured. However, it is important to acknowledge that the different forms of 2,4-D will have different physiochemical properties. Further measurements of 2,4-D are required to fully understand 2,4-D's volatilisation potential in the field through the measurement of a commercial products containing the different 2,4-D forms.  This is

particularly important as the CIMS observations suggest 2,4-D is more volatile than literature values suggest so is more likely to reside in the atmosphere and as such may impact environmental risk assessment.

This work highlighted differences in vapour pressure measurements compared to the literature values and has shown that for some compounds the observed values, have the potential to alter predicted impacts due to difference in gas-particle phase partitioning. However, the latter is only relevant if there is a mechanism by which the pesticide substance can volatilise either

during or after spraying. The importance of using the correct values of physiochemical properties in pesticide models have been  previously highlighted by (Couvidat et al., 2022) who included pesticides into the atmospheric chemical transport model CHIMERE.

It is recommended that future work expands on the range of pesticides studied using FIGAERO-CIMS to provide observed vapour pressure data for incorporation into realistic atmospheric transport and chemistry models and to improve the robustness

of environmental assessments of pesticide environmental impact via atmospheric pathways. It is also recognised that vapour pressure is just one section of a jigsaw to understand how a pesticide may be transported in the atmosphere. Further studies must also consider atmospheric reactions and secondary organic aerosol (SOA) formation in order to develop a fuller picture of the atmospheric fate and behaviour of the pesticide substances.

## 5   Summary

It has been demonstrated that the FIGAERO-CIMS provides a robust and comparable method of volatility measurement based on a well characterised set of calibration compounds, PEGs over a range of vapour pressures, incorporating those of atmospheric relevance. This approach has enabled determination of the vapour pressures of six exemplar pesticides.

The calibration of the FIGAERO-CIMS to determine vapour pressures from the maximum signal in the target compound on evaporation from the filter during a temperature ramp, or thermogram ($T_{max}$) replicated the atomisation approach described by

A.Ylisirniö, (Ylisirniö et al., 2021). The results of this present study validate those of the previous work and also demonstrate that earlier methods of syringing calibrant material onto the FIGAERO filter fails to take account of kinetic evaporation times of semi-volatile material and should not be used. The novel use of the PEG-400 mixture allows one pre-made bulk mixture

that covers a wider range of vapour pressures compared to previous vapour pressure calibrations of beyond atmospherically relevant vapour pressures.

Vapour pressures of a number of pesticides that were categorised as Intermediate to low volatility were then measured using the FIGAERO-CIMS. This paper report vapour pressures and C* values along with the associated measurement uncertainty. Comparisons of the measured values with current literature values and a model based on structural activity relationships, are broadly in agreement, suggesting, that the CIMS values to be an accurate alternative value to be used in future models. However, for some compounds substantial differences were demonstrated, some of which are substantial enough to have

significant implications for transport pathways through the atmosphere and hence potentially inform future development of regulatory compliance.

Future work should confirm the differences in pesticide vapour pressures measured using the FIGAERO-CIMS compared to the currently accepted literature values and extend the range of measurements over a wide range of pesticides. These values can then be used in environmental models and environmental assessments, to support cases of contamination via volatilisation,

a route that may not have been previously considered important for several compounds if literature values under-estimate compound vapour pressures.

**Author Contribution**

Experimental planning and design were performed by OJ, AV, TB and HC. Design and building of the calibration set up at Manchester was designed by OJ, AV and TB, experiments were conducted by OJ, Discussions of results occurred with OJ,

AV, GM and HC. Modelling was carried out by OJ, SOM and pesticide legislation was advised by DJ. Data was plotted and analysed by OJ. The manuscript was written by OJ. All authors reviewed the manuscript before publication.

**Competing Interests**

The contact author has declared that none of the authors has any competing interests

**Acknowledgements**

Funding was received by the EPRSC CDT in Aerosol Science and partly funded by Syngenta Ltd. AV also acknowledges the support of the TOX-PEST project (NE/X010198/1).



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
