# Peer review of "Determination of the Atmospheric Volatility of Pesticides using FIGAERO - Chemical Ionisation Mass Spectrometry"

_EGUsphere, 2024_

## Author Comment (AC1)

**Responses to reviewer's #2 comments**

Jackson et al, present vapour pressures for six pesticides measured using the FIGAERO-ToF-CIMS analytical technique. There is a large discussion around the method of delivery of the pesticides onto the filter contrasting direct injection and atomisation. The method of calibration against a series of PEG compounds with known vapour pressures is also presented. The measured values are also contrasted with literature values and two SAR models.

This manuscript is of excellent scientific significance as it demonstrates the utility of the FIGAERO-ToF-CIMS measurement technique for investigating the volatility of -pesticides with high environmental relevance. This manuscript should be considered for publication after considering several major and minor comments.

We'd like to thank Referee 2 for their positive comments and to respond to the general and detailed comments as follows (reviewer comments in black and our responses in blue; the line numbers referred throughout are referring to the original manuscript). A marked-up version of the manuscript detailing the amendments from all of the reviewers comments is also provided.

Major comments

The manuscript is extremely comprehensive regarding background, summarising the current state of knowledge and where these new finding fit in a literature and regulatory context.

While informative, large parts of the text are quite verbose and do not seem entirely relevant to the focus of the manuscript. Additionally, throughout the manuscript and specifically in section 4.1, there seems to be a lot of repetition. In some instances, the text does not match the section it is found in, for example there is some description of the experimental method at the end of the SAR section. I found the discussion on comparisons with the literature quite hard to follow, mainly due to the lack of clarity around which 'literature' is used for comparisons.

That being said, these issues concern the presentation of the study, rather than the study itself. I think the manuscript would benefit greatly from editing down to focus on the key message(s) and better connecting the bigger picture issues of regulation and reporting to the results. More efficient organisation of the text would also greatly improve its focus. This would also give the author the opportunity to check some grammar issues and typos.

We thank Referee 2 for the major comment stated regarding the background information. This was also picked up in referee 3's comments. We are encouraged that the referee is positive about the thorough introduction and literature study being important, but we note that they request sharpening the text and reducing the verbosity. We will endeavour to retain the essence of the discussion in a revised version whilst sharpening the text and reducing it in size.

Minor comments

Line 87 – "effectively shielded from degradative gas phase oxidation". It would be nice to contrast this with condensed phase chemical processes. Is it possible to say something about the lifetime of the pesticide in the gas vs the condensed phase?

We thank the reviewer for pointing this out. Line 87 refers specifically to the pesticide difenoconazole which has been found to reside predominantly in the particle phase due to its vapour pressure of $3 \times 10^{-8}$ Pa, making it rather involatile in the atmosphere and thus would not be expected to reside in the gas phase for any significant timeframe.

However, with respect to the pesticides in this study estimations of condensed phase reactivity are relatively unknown due to the lack of understanding of many pesticides in the atmosphere environment (Brüggemann et al., 2024) and thus a more extensive experiment would be required to be undertaken, beyond the scope of the manuscript. With this the following has been added to the updated manuscript '*the very low vapour pressure of difenoconazole is making it practically involatile and thereby unable to react in the gas phase*'

Line 93 – what is meant by "activated process"?

We thank the reviewer for pointing this out. In our editing down and refocusing of the introduction this phrase has been removed from future versions of the manuscript.

Line 95 – what is meant by "pesticide active substance"?

We thank the reviewer for pointing this out. The pesticide active substance refers the active ingredient in the pesticide formulation i.e. the compound with the pesticidal properties. This is defined in line 40. To improve clarity the definition in line 40 has been edited to refer to definition as defining both pesticide active ingredients and pesticide active substances.

Line 125 (167) – what is meant by "reverse flushed"?

We thank the reviewer for highlighting this. This comment was also addressed by reviewer 1. Reverse flushed refers to the process within the operation of the FIGAERO desorption method in which the nitrogen flow is heated and flowed onto the filter containing the compound of- interest. In order to avoid confusion with the reader, the line now reads '*The filter (and any particles on it) is flushed with nitrogen and continuously heated at a rate of 8.75°C min$^{-1}$.*' This has been updated for further versions of the manuscript.

Line 146 – what "coefficient" is gamma?

We thank the reviewer for highlighting this. The sentence should have read $\gamma$ = condensed phase activity coefficient, which for pesticides is assumed to be 1. This has been updated for future versions of the manuscript.

Line 182 – flow of 20 Lm-1 of what gas?

We thank the reviewer for highlighting this. The gas referred to is purified compressed -air connected to a mass flow controller to control the flow of 20Lmin$^{-1}$. This has been edited into the manuscript.

Line 187 – Through -> although

We thank the reviewer for pointing this out. This has been corrected in the revised manuscript.

Line 199 - I found the description of SAR a little confusing and not introduced particularly clearly. For example, section 2.2 misses directly stating that SAR is used to predict vapor pressures, and there is no explanation of why the Nannoolal and MGM models are chosen specifically. I think this can be easily corrected by reformulating the text.

We thank the reviewer for pointing this out. This has been corrected in the revised manuscript to say the following '*Structure activity relationship models are commonly used as a first prediction and screening tool for a compound's physiochemical properties (including vapour pressure) and thus environmental fate and behaviour*'

Additionally, the clarity of the SAR method section has been adapted to reflect the reasoning behind the uses of the different vapour pressure estimation methods to say the following: '*The Nannoolal model was chosen because the vapour pressure observations used as training data for the*

*development of the model included a large number of aromatic compounds, with a wide range of functional groups'*

Line 200 – estimation methods of what?

This refers to the estimations of- physiochemical properties, specifically vapour pressure in this study.. We thank the reviewer for pointing this out and the manuscript has been revised: '*Estimation methods are required to predict the vapour pressure of atmospherically relevant compounds and are commonly achieved through equation-based estimation methods'*

Line 208 – "The Nannoolal model was chosen as training data for the model" doesn't really make sense.

We thank the reviewer for the comment, the new versions of the manuscript have been reworded to more clearly state the following – '*The Nannoolal model was chosen because, the vapour pressure observations used as training data for the development of the model included a large number of aromatic compounds, with a wide range of functional groups'.*

Line 263 – what is a PPP?

We thank the reviewer for the comment, PPP is plant protection product, this is defined in line 39 of the original manuscript.

Line 283 – the commercial PEG solution is a great benefit. Can you explain a bit more about why you are using PEG-4 as your lower Tmax limit? Although the atomisation method is only useful down to PEG-5, it doesn't look like the syringe method is great below that either? There is actually some discussion of this at line 338.

We thank the reviewer for this comment, a similar comment was raised by reviewer 1, the response to reviewer 1 a summary of the response is shown below:

The PEG calibration curve using the syringe method extends to PEGs 3 and 4 whereas that of the aerosol method ends at PEG5 due to the effects. In terms of the experiments carried out in this manuscript, the smaller sizes of the aerosolised PEG drops on the filter evaporate more rapidly at lower temperatures and hence cannot be measured by the FIGAERO method, whereas the larger syringed drops of the most volatile material remain present until larger Tmax values. The following has been added to the manuscript to increase clarity '*Here, the Tmax values from PEG3-8 are reported for the syringe method and PEG 5-8 for the atomisation method. The discrepancy between the two methods is due to the inability of the FIGAERO to measure the smaller more volatile droplets of PEG3 and 4 in the atomisation method'.*

Line 289 – I wonder if a 1:1 plot of atomisation vs syringe Tmax values for each PEG would give any insights into the systematic bias of the syringe technique?

We thank the reviewer for this comment, a detailed response to this issue has been provided in reviewer 1.

Line 304 – is it possible to say more about the impact of the 0.1 g/L vs 0.5 g/L deposition on the filter on the Tmax?

We thank the reviewer for the comment. We would like to direct the reviewer to the publication referenced in the text (A. Ylinsirniö et. al. 2021), which explores the differences between concentrations of particles on the filter. To add clarity, the following was added to the manuscript: '*It was suggested that an increase in concentration on the filter leads to higher Tmax values due to more energy required to evaporate off the filter. For the atomisation method, equation 3 was introduced to*

**Commented [HC1]:** Copy alterations/suggestions in here

*monitor the time required for the required mass to be deposited on the filter. This varies due to the output of the atomiser and is measured using the SMPS-CPC. Here, the integrated mass over the duration of the sampling period provided the total mass collected.* '

Commented [OJ2]: Add text to manusxript

Line 315 – "small but consistent repeatable effect". This isn't explained well here but I think this explained more fully on page 17 later. This is a good example of repeated information in two different places. It would be good to summarise this information in one place – maybe you can make a comment on the variability of particle diameter and how it affects Tmax in this instance – is it possible to have a metric like x degrees C / nm? Is the variation here significant?

We thank the reviewer for the comment and in the updated manuscript we have aimed to keep the information clear. The small but consistent repeatable effect refers to the effect due to the differing size to volume ratios of different sized particles found by (Ylisirniö et al., 2021). However, this exact value should not be used for the experiments presented here as different FIGAERO-CIMS should not be compared against when considering exact values. Despite this the change in Tmax of 7°C between particle sizes of 80 and 300nm determined in the previous publication can be considered as significant as it may lead to differing vapour pressures, to mitigate this, as suggested in this manuscript particle sizes must be kept the same in the same investigation.

A similar comment was made by reviewer 1, in which a more in-depth explanation is provided. We have included a statement in this section to make this point more clearly: '*This is explained by the increase in particle sizes used in the nebulisation of the calibration particles in this work compared to the previous work which had a smaller modal diameter of 60 nm (compared to the 105nm in this study). However, it must also be noted that it would be inappropriate to compare the exact measurements between the two non-calibrated FIGAERO set ups, instead it is important to recreate the size distribution within the same investigation*'.

Commented [HC3]: This answer needs to be much more brief and also add clarity. The general reader nor the editor will have any understanding of either the comment or your answer. I suggest Typing out the original sentence to give context and give a brief specific answer and refer to ref 1 answers more generally

Line 357 – "... highest tmax was for the least volatile pesticide .." least volatile according to who? Are you referencing a literature value or the fact that the tmax is highest for mesotrione?

We thank the reviewer for the comment, By definition the compound with the lowest Tmax will be the most volatile according to numerous sources including the Tmax values calculated by the FIGAERO as shown by the calibration curve in figure 3b and calculated by equation 2. To make this clearer, we added in the main text 'The raw thermograms (fig. 6) and eq. 2 were used to calculate the $T_{max}$ values presented in figure 7, where the highest $T_{max}$ was for the least volatile pesticide…"

Commented [AV4]: As I mentioned in the comment above, I think you can provide some more direct answers to the reviewer's queries. It will make a lot easier the review for them and less likely to get pissed off that you dodged the question or not properly replied.

Commented [OJ5]: Finish and add to text

Line 364 – Figure 8. These need an (a) and (b). Which Literature Value is being referred to in the legend? It would be better if the legend handle was a bit more descriptive.

We thank the reviewer for pointing this out. This has been corrected in future versions of the manuscript.

Line 376 – what is a "regulatory endpoint value"?

The "regulatory endpoint value" refers to the 'endpoints' defined in line 112 of the original manuscript in which the regulatory body it is referring to is EFSA (European Food Safety Authority).

Line 382 – MGM vs MGM SAR these refer to the same thing but are given different names.

Thank you for pointing out a possible confusion the SAR refers the MGM in being a Structure activity relationship model as defined in section 2.2 of the methods and thus is two acronyms, both previously defined in the text. To avoid any further confusion the 'SAR' has been removed from the phrase in future versions of the manuscript.

Line 385 – what is meant by comparison data? It would be better to be more specific with the literature value you are comparing to.

We thank the reviewer for the comment. The 'comparison data' refers to all the measured and modelled values compared for the Mesotrione pesticide in figure 8a. To avoid confusion, the future version of the manuscript has been changed so that the phrase reads: Mesotrione exhibits the greatest discrepancy between the measured values and the modelled values'

Line 386 – The University of Hertfordshire Pesticides Properties Database for Mesotrione is mentioned here, and then two lines later, the EFSA endpoints, from which the database draws its data. This section reads very narratively which is confusing, i.e., I am not sure it is necessary to explain that the database had a suspect value but on further inspection it was because the underlying EFSA data is measured at a different temperature. This just highlights an issue with the database.

It might be easier to standardise the way you refer to the different literature data earlier on to make this easier to follow. Is the "upper limit" value the green "stated literature" value in figure 9? Using these different terms is hard to follow.

We thank the reviewer for the comment and apologise for the confusions. In addition to other reviewers comments we have improved the clarity of this paragraph. It now reads:

*Previous studies of the vapour pressure value for Mesotrione were determined at 373K of $5.7 \times 10^{-6}$ Pa (Lewis et al., 2016) and so have been previously considered to be an upper limit value in regulatory framework literature since lower vapour pressures will occur at lower temperatures. We calculate the vapour pressure at 293K based on earlier work at higher temperatures (Efsa, 2016) using a Clausius-Clapeyron relationship and compare it with our observation in figure 9 along with MGM model predictions and show consistency across both observations and models for Mesotrione.*

*The observations at ambient temperatures presented in this work show Mesotrione to be considerably less volatile than may be expected based on the upper limit value, . It can be seen that the reason for the difference in the EFSA document (Efsa, 2016) is that the method used observed some thermal decomposition of Mesotrione (the melting point temperature is stated to be ~165°C/438K "with some decomposition on melting). However, Mesotrione can still be concluded as involatile as if a pesticide at the higher temperature (100°C) is non-volatile it will not be more volatile at 25°C and thus extrapolation to a lower temperature is not required.*

Line 394 – I am not entirely sure why thermal decomposition is mentioned here. What does difference in the EFSA document mean?

We thank the reviewer for the comment. the literature lacks detail on the reasons for this and so this sentence will be removed in the revised version of the manuscript.

Line 405 – Figure 9. What are the different "literature measurements at varying temperatures"? where do they come from? It would be more informative to have the sources rather than stating they are measured at various temperatures (this is what the x axis shows already). The 'stated literature value' is just plotted at the wrong 1/T value. If measured at 373K then this should be plotted at 1/373 = 0.000268 and so would appear to follow the trend.

We thank the reviewer for this comment. The placing of the literature value was meant to highlight where the literature value would be placed on the Clausius Clapeyron graph if the stated literature value was taken at 20°C and not the 100°C it was actually determined at. However, we recognise this was confusing so the plot has been replotted (as shown below) to show the true temperature at which the literature value was determined, in agreement with this comment the point now aligns with the trend.

Commented [HC6]: Ensure this includes my suggested changes to ref 1 response.

[Figure]

In addition, the following clarification has been adapted to the manuscript to further explain how figure 9 was plotted: '*Figure 9 plots each of the measured values  (at 323K, 373K, 383K) taken from EFSA's list of  endpoints and compares these values to the FIGAERO-CIMS measurement and the MGM predicted value*.' Additionally, the caption to figure 9 now states: ' *Figure 9 Clausius Clapeyron plot of Mesotrione measurements from the EFSA endpoint report (Efsa, 2016)  determined at varied temperatures (323K, 373K, 383K). This is then ­compared with the CIMS atomisation measurement and the Modified Grain Method (MGM). Points in blue are the values from the EFSA endpoints Efsa, 2016*).'

Line 418 – "within an order of magnitude". It is of course good the difference in values is less than an order of magnitude, but without uncertainty measurements it is difficult to assess how 'good' the agreement is. Either a measure of the variability or uncertainty.

We thank the reviewer for the comment. Figure 8a contains error bars for the CIMS measurements, this has been highlighted in the legend/caption of the figure. However, are small enough due to the high repeatability of the Tmax in these experiments.  The following has been added to the figure 8 caption ' *The values come from the average of 3 runs shown in figure 7. The error bars are shown on the figure however are not visible due to the high repeatability.*' On the other hand, a comparison of the accuracy of each of the methods has not performed in previous studies and thus a full conclusion may be inappropriate. A study using a number of different compounds not just pesticides would be required to assess the performance.  This is especially important when considering the comparisons with the models. This is beyond the scope of the manuscript.

Line 420 – Do you expect higher volatility with more functionalisation? Can you explain further?

We would like to thank the reviewer for the comment. On reconsideration of the trend we have consequently, replaced this sentence with the following, which gives a more meaningful and insightful interpretation of the estimation methods.  '*Overall, both models may lead to misleading indications of the environmental fate of a pesticide. This is because the MGM, and to a lesser extent Nannoolal, predicts likely gas-phase presence of 2,4-D, dicamba, MCPA and MCPP, whereas measurement indicates a much stronger tendency to be present in the particle phase, or to remain on the target application*.'

Line 456 – "Lower volatility compounds are unlikely to be atmospherically relevant if applied to the surface in the liquid or solid phase". I take it this refers to application of pesticide to the surface of a plant? Low volatility compounds are of course atmospherically relevant in an aerosol context.

We thank the reviewer for the comment and agree that the message of the last sentence is confusing and may be irrelevant for this section. Of course, low volatility compounds are atmospherically important in the an aerosol context. However, from legislation perspective low volatility compounds present minor concern as they do not expect to be important for long-range transport. This has been picked up by another reviewer (see below) and we have made an effort to be clearer about the context of our statements. In the light of this, the authors have decided to remove the last two sentences to aid with clarity.

The following comment was also made by reviewer 1. A more in depth response and the detail of amendments can be found here.

Line 457 – I don't understand the message of the last sentence of this paragraph.

We thank the reviewer for the comment and agree that the message of the last sentence is confusing and may be irrelevant for this section. The authors have decided to remove the last two sentences to aid with clarity.

Line 467 - "training dataset" what data does this refer to? This is the Nannool training data? I think this just requires more consistency in naming.

We thank the reviewer for the comment. 'Training data' refers to training data sets used for both the MGM and Nanoolal models. This has been made clearer for future manuscript versions and reads '…*the chlorinated benzene or carboxylic acid function groups may have been poorly represented in the training data set used to develop the MGM and Nannoolal models'*

Line 470 – "fishtine factor" needs introducing earlier in the text.

We thank the reviewer for the comment, The fishtine factor refers to the factor used in the calculation of the model, however on further review the authors believe that is inappropriate to include in the context of the Nannoolal model. Instead, a general comment was added to allow further comparison. This was: 'For four of the five pesticides where both estimation methods were applied, the Nannoolal method gave better agreement with observed vapour pressures, consistent with the findings of Barley and McFiggans (2010) (which included multi-functional aromatics).'

Line 490 – " … volatilisation is a major degradation mechanism .. " volatilisation doesn't degrade the active ingredient, I guess this means degradation of the pesticide product itself?

We thank the reviewer for the comment, the manuscript has been altered to be correctly phrased: '…volatilisation is a major mechanism of loss from the initial application site.'

Brüggemann, M., Mayer, S., Brown, D., Terry, A., Rüdiger, J., and Hoffmann, T.: Measuring pesticides in the atmosphere: current status, emerging trends and future perspectives, Environmental Sciences Europe, 36, 10.1186/s12302-024-00870-4, 2024.
EFSA: Peer review of the pesticide risk assessment of the active substance mesotrione, EFSA Journal, 14, 10.2903/j.efsa.2016.4419, 2016.
Lewis, K. A., Tzilivakis, J., Warner, D. J., and Green, A.: An international database for pesticide risk assessments and management, Human and Ecological Risk Assessment: An International Journal, 22, 1050-1064, 10.1080/10807039.2015.1133242, 2016.
Schobesberger, S., D'Ambro, E. L., Lopez-Hilfiker, F. D., Mohr, C., and Thornton, J. A.: A model framework to retrieve thermodynamic and kinetic properties of organic aerosol from compositionresolved thermal desorption measurements, Atmospheric Chemistry and Physics, 18, 14757-14785, 10.5194/acp-18-14757-2018, 2018.

Ylisirniö, A., Barreira, L. M. F., Pullinen, I., Buchholz, A., Jayne, J., Krechmer, J. E., Worsnop, D. R., Virtanen, A., and Schobesberger, S.: On the calibration of FIGAERO-ToF-CIMS: importance and impact of calibrant delivery for the particle-phase calibration, Atmospheric Measurement Techniques, 14, 355-367, 10.5194/amt-14-355-2021, 2021.

---

## Author Comment (AC2)

**Responses to reviewer's #1 comments**

**General Comments**

Jackson et al. provides volatility measurements for a set of 6 pesticides in current use through application of a filter desorption method utilizing the Filter Inlet for Gases and AEROsols coupled to an iodide mode Chemical Ionization Mass Spectrometer (FIGAERO-CIMS). The authors discuss broad variability and inconsistency in measured and reported vapor pressures (volatility) for many compounds, dependent on measurement method. FIGAERO-CIMS volatility is derived from a relationship between temperature of maximum thermal desorption signal (Tmax) and volatility (increasing Tmax indicates lower volatility). This relationship is quantified by calibration of the technique against a set of polyethylene glycol (PEG) polymers of different lengths, which can be loaded onto the filter by syringe deposition or atomization. This calibration is then utilized to assess the volatility of the target pesticides when they are atomized in a mixture or individually.

Jackson et al. conclude that, in agreement with prior literature, use of a syringe for calibrant or sample delivery is less effective than atomization. However, as noted in specific comments below, greater consistency is achieved between literature PEG Tmax values using the syringe method than atomization. While arguments from prior literature are used to highlight the unsuitability of the syringe method, the data as presented does not provide clarity as to why syringe delivery is unacceptable. Additional quantification of thermogram peak width and subsequent resolution or overall variability of each measurement method would provide valuable insight into the failings of the syringe method for PEG calibration.

We'd like to thank Referee 1 for their positive comments and to respond to the general and detailed comments as follows (reviewer comments in black and our responses in blue; the line numbers referred throughout are referring to the original manuscript). An updated version of the manuscript detailing the amendments from all of the reviewers comments is also provided.

In response to the general comment, the authors would like to clarify the argument presented in this paper regarding the syringe and atomisation method. It is not that the syringe an unacceptable method. Instead, we present the atomisation method as a more suitable method to be used in this study due to the following. Most importantly, the size of the calibration droplets on the FIGAERO filter need to be similar to those of the material being sampled for the calibration to be representative, in order to control this the atomisation method is required, in which a number of small droplets are deposited on the filter, compared to one large droplet when the syringe method is used. As pointed out by Ylisirnio et al. this is due to complete evaporation of a single compound of a given volatility from the filter during the thermal ramp being dependent on the size of the particle (due to varying surface-volume ratio). A molecule in the larger droplet from the syringe method takes more energy to evaporate in comparison to the smaller droplets on the filter in the atomisation method and therefore full evaporation takes longer. This results in the Tmax compound of the same volatility being higher in the syringe method and the overall thermograms being broader (as observed in figure 2 and backed up by the gaussian fits provided in S1). Secondly, the atomisation method presents a more relevant method compared to online sampling in the field and thus provides an opportunity to compare against results if required. Finally, the syringe method delivers a larger volume of material and this risks contamination of the mass spectrometer – see later discussion of the peak shapes. Overall, the atomisation method is the optimum method for sample delivery to the FIGAERO. In the amendments throughout the paper, we hope the reviewer is able to further appreciate the requirement for this calibration . To address this in the paper the following has been added to the end of the PEG discussion section (4.1)as an overview of the section: ' *Overall, the atomisation method was chosen since it was recommended in the previous literature and shown (S1 and S2) to be more repeatable and*

*representative than the syringe method and lead to far less low volatility material entering the mass spectrometer, reducing the contamination'*

Additionally, we must make the reviewer aware that it is only suitable to compare the data from the same apparatus and the particles measured are the same size. In addition, this means that any calibration must have the same particle size as the compounds of interest (e.g pesticides) as the size of the compound impacts the volatility in the atomisation method.

Repeats of thermograms of representative PEGs covering the atmospherically relevant volatility range are presented and provided in the supplementary as part of the gaussian fits which is accompanied by the standard deviation and full width half maximum of each peak, this aims to provide clarity on the repeatability of the experiments and the gaussian fit of the thermograms. An example of the summary tables provided in the supplementary material is shown below:

**Supplementary Table 1: Presented standard deviation and full width half maximums for each of the pesticide atomisation thermograms.**

| Compound | Run Number | Standard Deviation | Full Width Half Maximum |
|---|---|---|---|
| 2,4-D | Pesticide run 1 | 0.187182 | 18.1 |
| Dicamba | Pesticide run 1 | 0.241627 | 16.3 |
| MCPA | Pesticide run 1 | 0.163907 | 15.9 |
| Mesotrione | Pesticide run 1 | 0.196694 | 18.5 |
| Mecoprop-P | Pesticide run 1 | 0.207007 | 20.6 |
| Trilfuralin | Pesticide run 1 | 0.202343 | 44 |
| 2,4-D | Pesticide run 2 | 0.074938 | 36.5 |
| Dicamba | Pesticide run 2 | 0.194977 | 58.9 |
| MCPA | Pesticide run 2 | 0.163907 | 16 |
| Mesotrione | Pesticide run 2 | 0.320658 | 41.3 |
| Mecoprop-P | Pesticide run 2 | 0.207007 | 21 |
| Trilfuralin | Pesticide run 2 | 0.271879 | n/a |

The above text gives an overview of the comments presented by the reviewer. More specific answers are provided below.

Pesticide volatilities are measured and reported for 6 pesticides of interest and compared to literature values, as well as values derived from 2 structure-activity relationship models. Here, Jackson et al. show close agreement between literature and FIGAERO-measured values for Dicambia, MCPA, and MCPP. Volatility of 2,4-D is measured as more than 1 order of magnitude higher than the literature value, explained here by differences in the PEG calibration particle size distribution. Volatility of mesotrione is measured as more than 2 orders of magnitude lower than the literature value, explained by error in the measurement of said literature value (measurement conducted at a higher temperature than this study). A Clausius-Clapeyron relationship is used to argue that observed and modeled volatilities are reasonable given the temperature difference between the two measurements. Trifluralin is noted as the highest volatility pesticide measured here. However, as noted in specific comments below, the data shown in Figure 6 seems to contradict this, since the Tmax observed there is substantially higher than the 27.7C reported in Figure 7. Furthermore, the desorption profile of Trifluralin has a uniquely large right tail and further assessment and analysis of this species behaviour would be appreciated.

We thank the reviewer for the comment. The plotting of the graph was revisited, and the incorrect temperature had been plotted, this has been amended and updated below. We greatly appreciate the referee spotting this error. The tail can be explained by the high concentration of pesticide injected onto the FIGAERO and sampled into the CIMS, resulting in the compounds coating the IMR and coming off over time. As Trifluralin is relatively volatile (compared to the other pesticides introduced into the CIMS), it will quickly re-evaporate, creating the tail. The other pesticides on the other hand will re-evaporate on timescales longer than that of a thermogram and so will be observed by an increase in background over the course of multiple experiments. At the start of each experiment, it was ensured that each of the pesticide signals returned to background levels before a new experiment was begun, as well as making sure the filter has properly cooled down to make sure no initial evaporation of Trifluralin occurs before the start of the experiment and thus the initial peak portion of the experiment is unaffected by the tail. To improve clarity of this issue a discussion should be added to a revised manuscript to say the following: '*The tail observed in the Trifluralin peak is due to the large concentration of pesticides injected into the mass spectrometer, leading to some residue being present in the IMR which can then re-evaporate creating a tail. This is not visible for the other compounds measured as they are of lower volatility and therefore would expect to remain on the walls of instrument for longer than the time taken for a single thermogram, potentially coming off in the background over time.*'

Generally, Jackson et al. concludes that FIGAERO-CIMS measurements of volatility are valuable additions to pesticide environmental assessment to understand the fate of toxic pesticides in the environment. Such measurements can improve modeling of fate and transport and environmental persistence. The manuscript presents a unique set of measurements of a particular set of pesticides as an example of this technique and its reasonable agreement with other vapor pressure measurements. Furthermore, the manuscript identifies a series of key concerns in the vapor pressure measurement space, including poor documentation of measurement methodologies including temperature measurement.

The manuscript would benefit from additional quantification when making comparisons between measurements and measurement techniques, both in terms of quantified Gaussian goodness of fit and peak shape metrics, and in terms of presenting uncertainty in the form of standard deviations or propagated error. Furthermore, additional literature comparisons where possible would be valuable. As discussed in the technical comments, small updates to the presented figures would improve legibility, consistency, and clarity across the manuscript.

Overall, the manuscript provides a unique measurement of pesticide volatility and is an informative first step toward building a more consistent picture of pesticide fate and transport in the environment. I appreciate the author's consideration of this reviewer's comments and their ongoing work to illuminate this important area.

We'd like to thank Referee 1 for their general comments and especially appreciation for the requirements needed for this work and understanding in the importance of this topic area of both FIGAERO-CIMS and pesticide research. Responses regarding the clarity and legibility of the manuscript will be addressed in the final version and the authors appreciate the specific and technical comments made by the referee, all of which have been taken into consideration.

**Specific Comments**

*Line 105-107:* The authors describe how volatility measurements may be conducted at higher temperatures and be extrapolated to lower temperatures. However, later in the manuscript when discussing this effect with mesotrione, it seems that rather than being extrapolated to lower temperatures, the measurements at high temperatures are simply used at low temperatures (there is no particular extrapolation). The authors then describe using a Clausius-Clapeyron relationship as an

appropriate adjustment to vapor pressure. How widespread is the practice of ascribing high temperature vapor pressure measurements to low temperature conditions?

We would like to thank the reviewer for pointing out the lack of clarity regarding this topic. The literature values used for Mesotrione were provided as reference data from a safety case perspective for the approval of the use of Mesotrione. The basis of the arguments in the case are that if the vapour pressure is sufficiently low at the higher temperatures under the measured conditions, then the volatility of the pesticide will be even less at the ambient temperatures and so will not undergo significant transport (this is central to regulatory compliance).  Rather than use the upper limit value that is applied for compliance purposes we have used a Clausius-Clapeyron approach to extrapolating the volatility to lower temperatures to show that the extrapolated value is consistent with our observations/. To improve the clarity of this the following text should be added to a revised version of the manuscript.

'*Previous studies of the vapour pressure value for mesotrione were determined at 373K of $5.7x10^{-6}$ Pa (Lewis et al., 2016) and so have been previously considered to be an upper limit value in regulatory framework literature (Efsa, 2016) since lower vapour pressures will occur at lower temperatures. We calculate the vapour pressure at 293K by extrapolating the Lewis et al data included in (Efsa, 2016) to 298K using a Clausius-Clapeyron relationship and compare it with our observation in figure 9 along with MGM model predictions and show consistency across both observations and models for Mesotrione.'*

*Line 147-152:* What are the implications of pesticides residing in a particular volatility class? What inference can you make about the fate and transport of that pesticide in the environment? Both here and elsewhere the authors comment on the volatility classes, but I think additional clarity in interpreting these classes would be valuable.

There are a number of implications of the volatilities of pesticides. The manuscript has been edited to state the following: '*The consequence of a pesticide having a higher volatility gives rise to the higher likelihood of a pesticide residing predominantly in the gas phase and thus less likely to undergo wet or dry deposition than a pesticide of lower volatility in the particle phase.*' Firstly, the phase at which it exists in in the atmosphere influences the potential for atmospheric transport or deposition, as well as the mechanisms of wet or dry deposition. If a pesticide is predominantly in the particle phase, then it is more likely to undergo deposition and less likely to be transported large distances from the source, whilst the opposite is more likely to be true if a pesticide is more volatile and thus more likely to exist in the gas phase. This understanding is required if we are to move to an atmospheric model of pesticide transfer through the air.

*Line 165:* Please note the thickness, pore size, and brand of the filter used in addition to the material. Similarly note purity and sources for all materials when described in the methods section.

 The filters used were 25mm diameter and 2.0µm pore size PTFE filters purchased from Cobetter lab. This will be amended in the manuscript.

*Line 167:* In what sense is the filter "backflushed"? The flow direction of heated nitrogen is the same as the initial particle sampling flow. (This may just be a filter sampling terminology I am unfamiliar with!)

We thank the reviewer for highlighting this. This comment was also addressed by reviewer 2.. Reverse flushed refers to the process within the operation of the FIGAERO desorption method in which the nitrogen flow is heated and flowed onto the filter containing the compound of  interest. In order to avoid confusion with the reader, a revised line now read '*The filter (and any particles on it) is*

*flushed with nitrogen and continuously heated at a rate of 8.75°C min⁻¹.'* This will be updated in further versions of the manuscript.

*Line 178-179:* Here the authors mention Bannan et al. (2019). I would suggest including PEG Tmax data from that study for comparison in addition to Ylisirnio et al. (2021). In general, I think the manuscript would benefit from a broader range of comparison to FIGAERO data sets where possible to better capture the variability in available measurements and the corresponding uncertainty in the measurement here.

The Bannan et al. (2019) paper was the first to use the PEG series to calibrate the FIGAERO CIMS Tmax data for volatility following on from the work of Krieger et al which details a range of methods to measure vapour pressure from PEG calibrations. However, the Bannan et al study used the syringe method, which Ylisrnio et al subsequently showed led to extended evaporation times of droplets on the filter. The important point here is that since the evaporation time of material of a given volatility from the filter is dependent on the size of the droplet on the filter the calibration curve that relates Tmax to volatility will be highly dependent on the droplet size as well as on the characteristics of the FIGAERO filter, the nitrogen flows and the temperature ramp rates. There is therefore no meaning in comparing Tmax v volatility calibrations from different instruments under different conditions with different sized droplets on the FIGAERO filter. We would go further and recommend that in order to derive volatility from the FIGAERO CIMS the calibration must be performed using the same system under the same conditions by depositing calibration particles with the same size distribution as the particles whose volatilities are to be determined.

*Line 190:* Was precisely 1 ug of atomized particles deposited on the filter for each measurement in this study based on live SMPS data?

We thank the reviewer for the comment. The 1ug of atomised filter was measured using equation 3 where $c_{SMPS}$ is the average concentration measured by SMPS data during the experiment. The following has been added to an updated manuscript '*to ensure that 1µg mass was delivered onto the filter during each experiment, the sample time was determined using equation 3 by monitoring $c_{SMPS}$.during the course of each experiment*'

*Repeatability:* Were calibrations and measurements repeated in this study? I see that calibration measurements were repeated 3 times in the caption of figure 3. Please add to the body text as well. What is the distribution of measurements, and can uncertainty be represented by error bars on many of the figures here?

We thank the reviewer for the comment. Figure 3 contains error bars; however, the calculated errors are small and not particularly visible in the figure. The text has been updated to include this in the main body of the text. In addition to this, several runs have been added to the supplementary data to explore the repeatability of the thermograms which are accompanied by tables which include the standard deviation and full width half maximum of each thermogram (shown below in supplementary figure 1). From this we can see that the repeatability is good across the experiments in the atmospherically relevant portion.

**Supplementary Table 2: Presented standard deviation and full width half maximums for each of the PEG syringe thermograms**

| Compound | Run Number | Standard Deviation | Full Width Half Maximum |
|---|---|---|---|
| PEG 1 | PEG Syringe 1 | 0.043029 | n/a |
| PEG 2 | PEG Syringe 1 | 0.019841 | n/a |
| PEG 3 | PEG Syringe 1 | 0.061816 | n/a |
| PEG 4 | PEG Syringe 1 | 0.120898 | n/a |

| | | | |
|---|---|---|---|
| PEG 5 | PEG Syringe 1 | 0.141968 | 63.7 |
| PEG 6 | PEG Syringe 1 | 0.213258 | 652.4 |
| PEG 7 | PEG Syringe 1 | 0.249988 | 49.4 |
| PEG 8 | PEG Syringe 1 | 0.255238 | 46.9 |
| PEG 9 | PEG Syringe 1 | 0.267166 | 46 |
| PEG 10 | PEG Syringe 1 | 0.279353 | 45.3 |
| PEG 11 | PEG Syringe 1 | 0.279771 | 42.5 |
| PEG 12 | PEG Syringe 1 | 0.274312 | 42.1 |
| PEG 13 | PEG Syringe 1 | 0.264753 | 40.9 |
| PEG 14 | PEG Syringe 1 | 0.259454 | 40.7 |
| PEG 15 | PEG Syringe 1 | 0.227712 | 38.7 |
| PEG 16 | PEG Syringe 1 | 0.152519 | n/a |
| PEG 1 | PEG Syringe 2 | 0.055531 | n/a |
| PEG 2 | PEG Syringe 2 | 0.012884 | n/a |
| PEG 3 | PEG Syringe 2 | 0.070254 | n/a |
| PEG 4 | PEG Syringe 2 | 0.159831 | n/a |
| PEG 5 | PEG Syringe 2 | 0.158666 | 58.9 |
| PEG 6 | PEG Syringe 2 | 0.226129 | 51.6 |
| PEG 7 | PEG Syringe 2 | 0.287126 | 51.6 |
| PEG 8 | PEG Syringe 2 | 0.303555 | 49.4 |
| PEG 9 | PEG Syringe 2 | 0.307325 | 47.7 |
| PEG 10 | PEG Syringe 2 | 0.319395 | 48.3 |
| PEG 11 | PEG Syringe 2 | 0.329424 | 47.5 |
| PEG 12 | PEG Syringe 2 | 0.315942 | n/a |
| PEG 13 | PEG Syringe 2 | 0.278748 | n/a |
| PEG 14 | PEG Syringe 2 | 0.22262 | n/a |
| PEG 15 | PEG Syringe 2 | 0.150391 | n/a |
| PEG 16 | PEG Syringe 2 | 0.069654 | n/a |
| PEG 1 | PEG Syringe 3 | 0.080361 | n/a |
| PEG 2 | PEG Syringe 3 | 0.039363 | n/a |
| PEG 3 | PEG Syringe 3 | 0.140204 | n/a |
| PEG 4 | PEG Syringe 3 | 0.231058 | n/a |
| PEG 5 | PEG Syringe 3 | 0.22881 | 34.9 |
| PEG 6 | PEG Syringe 3 | 0.219188 | 32.4 |
| PEG 7 | PEG Syringe 3 | 0.24527 | 32.6 |
| PEG 8 | PEG Syringe 3 | 0.253734 | 35 |
| PEG 9 | PEG Syringe 3 | 0.276235 | 34.3 |
| PEG 10 | PEG Syringe 3 | 0.285409 | 34.2 |
| PEG 11 | PEG Syringe 3 | 0.291096 | 31.2 |
| PEG 12 | PEG Syringe 3 | 0.242444 | 25.5 |
| PEG 13 | PEG Syringe 3 | 0.158333 | n/a |
| PEG 14 | PEG Syringe 3 | 0.203823 | n/a |
| PEG 15 | PEG Syringe 3 | 0.064268 | n/a |
| PEG 16 | PEG Syringe 3 | 0.033518 | n/a |

**Supplementary Table 3: Presented standard deviation and full width half maximums for each of the PEG atomisation thermograms**

| Compound | Run Number | Standard Deviation | Full Width Half Maximum |
|---|---|---|---|
| PEG 1 | PEG Atomisation run 1 | 0.003731 | n/a |
| PEG 2 | PEG Atomisation run 1 | 0.027495 | n/a |
| PEG 3 | PEG Atomisation run 1 | 0.173071 | n/a |
| PEG 4 | PEG Atomisation run 1 | 0.063222 | n/a |
| PEG 5 | PEG Atomisation run 1 | 0.188054 | n/a |
| PEG 6 | PEG Atomisation run 1 | 0.214379 | 31.6 |
| PEG 7 | PEG Atomisation run 1 | 0.223407 | 31.7 |
| PEG 8 | PEG Atomisation run 1 | 0.22717 | 31.7 |
| PEG 9 | PEG Atomisation run 1 | 0.247393 | 35.1 |
| PEG 10 | PEG Atomisation run 1 | 0.277645 | 42.5 |
| PEG 11 | PEG Atomisation run 1 | 0.308204 | 51.2 |
| PEG 12 | PEG Atomisation run 1 | 0.298668 | 50.5 |
| PEG 13 | PEG Atomisation run 1 | 0.249717 | 45.4 |
| PEG 14 | PEG Atomisation run 1 | 0.183889 | 43 |
| PEG 15 | PEG Atomisation run 1 | 0.089173 | 37.4 |
| PEG 16 | PEG Atomisation run 1 | 0.031918 | n/a |
| PEG 1 | PEG Atomisation run 2 | 0.065051 | n/a |
| PEG 2 | PEG Atomisation run 2 | 0.11932 | n/a |
| PEG 3 | PEG Atomisation run 2 | 0.134942 | n/a |
| PEG 4 | PEG Atomisation run 2 | 0.195675 | n/a |
| PEG 5 | PEG Atomisation run 2 | 0.170252 | 42.2 |
| PEG 6 | PEG Atomisation run 2 | 0.234296 | 36 |
| PEG 7 | PEG Atomisation run 2 | 0.241734 | 33.4 |
| PEG 8 | PEG Atomisation run 2 | 0.216853 | 30 |
| PEG 9 | PEG Atomisation run 2 | 0.196929 | 30 |
| PEG 10 | PEG Atomisation run 2 | 0.181912 | 31.7 |
| PEG 11 | PEG Atomisation run 2 | 0.162213 | 35.8 |

| | | | |
|---|---|---|---|
| PEG 12 | PEG Atomisation run 2 | 0.15842 | 49.1 |
| PEG 13 | PEG Atomisation run 2 | 0.169465 | n/a |
| PEG 14 | PEG Atomisation run 2 | 0.149642 | n/a |
| PEG 15 | PEG Atomisation run 2 | 0.143715 | 51.7 |
| PEG 16 | PEG Atomisation run 2 | 0.172397 | 45.6 |
| PEG 1 | PEG Atomisation run 3 | 0.008783 | n/a |
| PEG 2 | PEG Atomisation run 3 | 0.051641 | n/a |
| PEG 3 | PEG Atomisation run 3 | 0.190573 | n/a |
| PEG 4 | PEG Atomisation run 3 | 0.033878 | n/a |
| PEG 5 | PEG Atomisation run 3 | 0.206966 | n/a |
| PEG 6 | PEG Atomisation run 3 | 0.203647 | 29.8 |
| PEG 7 | PEG Atomisation run 3 | 0.209143 | 28.7 |
| PEG 8 | PEG Atomisation run 3 | 0.213781 | 27.9 |
| PEG 9 | PEG Atomisation run 3 | 0.214382 | 28 |
| PEG 10 | PEG Atomisation run 3 | 0.221895 | 29.6 |
| PEG 11 | PEG Atomisation run 3 | 0.255236 | 34.5 |
| PEG 12 | PEG Atomisation run 3 | 0.311995 | 43.1 |
| PEG 13 | PEG Atomisation run 3 | 0.306441 | 41.2 |
| PEG 14 | PEG Atomisation run 3 | 0.209504 | n/a |
| PEG 15 | PEG Atomisation run 3 | 0.13016 | n/a |
| PEG 16 | PEG Atomisation run 3 | 0.06895 | n/a |

*Methodology:* Why were separate calibration and pesticide volatility desorptions necessary? Could PEG be included in the pesticide solutions and atomized simultaneously for a concurrent volatility calibration during pesticide desorption?

We thank the reviewer for the comment. Yes the PEG mix could be used and atomised simultaneously and it is appreciated that experimental biases involving differences in analyte and calibrant size distribution biases. In practice this increases complexity since PEGs may interact with the pesticides, the mass spectrum would be more complex. This also allowed the verification the volatility vs Tmax relationship in advance of conducting the pesticide experiments.

*Line 249-250, "Conversely it is also important to determine the volatility of pesticides thought to be involatile (i.e., no chance of volatilization in the atmosphere) to ensure that there is no potential for atmospheric presence thus no further risk assessment in air is required.":* Here, this argument is made the low volatility species have no atmospheric relevance. In other portions of the manuscript, the authors note that low volatility species can be sequestered in particles and avoid atmospheric oxidation and be transported long distances (Line 89) or that low volatility species can be resuspended in dirt or dust particles (Line 490). Is this line intended to capture the current state of environmental risk assessment policy? If so, that should be clear and distinct from the broader commentary the authors make on potential environmental fate based on the results of this study.

We thank the reviewer for their comment and agree that is trying to reflect the current state of the environment legislation. However, we would like to highlight that the phrase '*is no potential for atmospheric presence thus no further risk assessment in air is required*' refers to state of the EU legislation in which if a pesticide active ingredient is proven to be sufficiently involatile than no further atmospheric risk assessment is considered as the chance of volatilisation is negligible, thus the environmental and health risks due to atmospheric transport are low. Therefore we will adapt the manuscript to use the word negligible and clearly define that the phrases used are from the state of the legislation. Specifically the manuscript will be adapted to say the following 'as EU legislation *states that if a pesticide active ingredient is proven to be sufficiently involatile than no further atmospheric risk assessment is considered as the chance of volatilisation is negligible, thus the environmental and health risks due to atmospheric transport is low (Regulation (EC) No. 1107/2009)*'

*Line 283-286:* In this discussion of thermogram shape and uniformity, some quantification of the PEG desorption curves may be valuable. What is their goodness of fit to a gaussian and their full width at half maximum?

We thank the reviewer for the comment and encouragement to delve deeper in to the statistics, these values are available be viewed in the supplementary material with the availability of the full width half maximum and standard deviation which has been added to earlier responses to the reviewer. In addition, a gaussian fit was produced for each of the thermograms and has been provided in the supplementary figures of the manuscript (S1). This was produced through mirroring the calculated gaussian (which was calculated using the $T_{max}$, full width half maximum and standard deviations). The idea is the deviation from gaussian can give information on how the tail impacts the plot.

The following has been added into the manuscript to support this information:

'*This observation is backed up by the gaussian fitted peaks shown in S1 (and complimentary statistics in S2) here a gaussian fit performed in the low side of the curve and mirrored (using the $T_{max}$ as the mirror line), assessing the gaussian shape of the thermogram which shows how the plot deviates from gaussian after the $T_{max}$ and therefore is not impacted by the tail before the $T_{max}$.*'

[Figure]

**S1: Repeat run fitted with a low T-side gaussian fit with a mirrored high T-side fit from the PEG syringe thermograms presented in figure 6 of the main manuscript.**

*Figure 3a:* Comparison between Ylisirnio et al (2021) using both methods here shows the increased Tmax associated with the syringe method over the atomization method which is well discussed here. However, Tmax data seems to be more consistent between the syringe method in both studies, with a bigger gap in Tmax observed between Ylisirnio and this study when atomization is used. Given that repeatability of vapor pressure measurements and consistency across measurement types is so poor, could you further explain/discuss if this apparent consistency when using the syringe method is desirable for vapor pressure estimation in targeted measurements?

The PEG calibration links the Tmax to the known vapour pressure of the PEGs. However, the process by which the FIGAERO is heated means that the evaporation rates of compounds from the filter surface will vary depending on the size of the liquid drop on the filter. A single large drop applied by

a syringe will take longer to evaporate at a given temperature compared to many smaller drops containing the same mass. Hence the calibration material needs to be applied to the FIGAERO surface in the same way as that of the analyte. This results in the atomisation method being preferred, as discussed at the beginning of reviewer 1's comments which is supported by the thermogram repeats in the supplementary material. As we discuss in the paper, our aerosol distribution applied to the filter surface was larger than that used in Ylisirnio et al and so our Tmax values are larger than theirs and exact comparisons between FIGAERO systems is not possible without ensuring the calibration particle sizes introduced onto the FIGAERO are similar. The repeatability of the syringing method is larger since it is harder to accurately replicate the droplet size introduced onto the filter. On the other hand when considering the comparison of the atomisation method, different (larger) droplet size distribution have been used in this study compared to that of Ylisirnio et al and so our Tmax v volatility relationship is shifted relative to theirs (further towards the syringe data). This results in the two calibrations not being comparable. Each set of calibrations are representative for the particular experiments since the aerosol size distributions of the investigated particles, in our case pesticides, were similar to those of the PEG calibration particles (figure 4). As a result, we recommend that the calibration particles used to derive the Tmax v volatility relationship are as close as possible to those of the particles whose volatilities are to be determined, as did Ylisirnio et al.

This is the case in figure 4 for all pesticides used except for 2,4 D and this is discussed in the results section. We have modified the text to make these points more clearly by adding the following text: '*Given the approach is subject to operational uncertainty we follow the recommendation of (Ylisirniö et al., 2021) and use the aerosolization method to determine pesticide vapour pressures on the basis that the atomisation method has a more robust repeatability and is more similar to atmospheric sampling conditions'*.

*Figure 3b:* There are two more points represented in the syringe data set than in the atomization data set. Why are these measurements not represented in Figure 5 (where both methods begin showing data at PEG-5)?

The PEG calibration curve using the syringe method extends to PEGs 3 and 4 whereas that of the aerosol method ends at PEG5 due to the effects discussed in the previous point. The smaller sized particles of the aerosolised PEG drops evaporate more rapidly from the filter than the larger syringed drops. The most volatile PEGs evaporate so rapidly from the small aerosolised drops that a Tmax cannot be resolved even at the lost temperatures of the ramp whereas the slower evaporation time of the larger syringed droplets means a peak can be resolved for the more volatile components.. The following will be added to the manuscript to increase clarity. '*Here, the Tmax values from PEG3-8 are reported for the syringe method and PEG 5-8 for the atomisation method, consistent with the available literature values presented by Krieger et al . As Kreiger et al state, this range covers all atmospherically relevant compounds that partition between gas and particle phases. As a result, while we can demonstrate that our approach to determining the Tmax of PEGs with the aerosol method can extend to larger PEGs we are unable to obtain a vapour pressure curve for these low volatilities at this stage. This analysis also demonstrates that since our thermograms closely resemble Gaussian distributions for PEGs 4 to 9 our results are representative across the whole range of relevant vapour pressures. Furthermore, vapour pressure values are only possible when injecting with the syringe due to the inability of the FIGAERO to measure the smaller more volatile droplets of PEG3 and 4 in the atomisation method due to rapid evaporation of the smaller droplets. Whilst beyond PEG8 the droplets are involatile and thus not atmospherically relevant*.'

*Line 313-314:* What differences are there between the size distributions used in this study and Ylisirnio et al (2021)?

The mode diameter of the PEG values reported in (Ylisirniö et al., 2021) are 60nm whilst the modal diameter of the PEGs in this study were 105nm. This statement will be addressed further in the reviewer's next comment.

*Line 320-321 and Figure 4:* Please provide some additional quantitative discussion of the particle size distributions observed (mode, total mass, spread, etc.) and if they are sufficiently similar to the PEG distribution for comparison and calibration per Ylisirnio et al (2021) as mentioned.

The referee is correct, this is the crux of many of the points raised in the discussion, which we hope we have clarified. The particle size distribution reported in (Ylisirniö et al., 2021) has a mode distribution value of 60nm. We have included a statement in this section to make this point more clearly: '*This is explained by the larger particle sizes used in the nebulisation of the calibration particles in this work (mode diameter~105 nm) compared to the previous work which had a smaller modal diameter of 60 nm. It is therefore not possible to directly compare the calibration curves but since the particles under investigation in both studies are similar in size to the calibration particles both calibrations can be effectively applied to the relevant experiments/. It is important to recreate the size distribution within the same investigation*'.

*Figure 4:* What density is assumed when calculated particle mass from SMPS particle volume distributions?

The density is assumed as 1000 $Kgm^{-3}$ thus assuming uniform standard density.

*Figure 6:* The shape of the pesticide desorption peaks seems, at least by eye, much sharper than the PEG desorption peaks. Can you provide quantitative information about peak shape statistics? How closely do the pesticides follow a Gaussian desorption curve?

A table has been provided below and will be included in the supplementary of a revised paper that includes the full width half maximum and standard deviation of each of the thermograms presented in this study. This suggests that the FWH maximum of the pesticides is predominately lower than in the PEGs. Additionally, we are able to compare the gaussian fits of the PEG calibration (shown in an earlier comment) with the pesticide gaussian fits shown below. Here it can be seen that all of the pesticides fit the gaussian peak well and that any deviation from gaussian occurs after the Tmax and therefore is not impacted by the tail before the Tmax.

[Figure]

**S3: Gaussian fitted pesticide thermograms presented in figure 6 of the main manuscript.**

*Figure 6 and Figure 7 and associated trifluralin discussion and conclusions:* The thermogram shape of trifluralin in particular is unique and worth discussion. It sharply appears at near 30 degrees and then peaks with a long tail to the right. Is this evidence of excessive trifluralin loading? Some type of thermal decomposition product? Furthermore, the Tmax for trifluralin in Figure 7 does not look correct based on Figure 6. There, the Tmax appears closer to 45 or 50 C, not the 27.7C reported. Please assess and correct as needed.

The Tmax of Trifluralin was replotted and the correct x axis temperature was used, and thus the 27.7°C is the correct value. This has been addressed in a previous point.

*Line 363-365:* This brief discussion of the potential bias in vapor pressure measurement of 2,4-D here is useful. Is there any method for extending this potential bias quantitatively to capture the anticipated error or uncertainty in the FIGAERO-CIMS volatility measurement?

We agree with the referee that the discussion of potential bias of 2,4-D  is useful because the size distribution is shifted to smaller sizes compared to the other experiments and the PEG calibration particles. The peak in distributions of the PEGs and most of the pesticides is 105 nm whereas that of 2,4 D is around 90 nm, 15 nm less. The Ylisirnio calibration used a distribution of particles with a peak at 60 nm and shows approximately the same Tmax for PEG6 as we show for PEG5 (see figure 3). In fig 3b a 15°C shift (a shift in PEG6 to PEG5) leads to a 3 orders of magnitude change in volatility. Since we see a 15 nm size shift for 2,4D compared to a 45 nm shift between the value presented here and Ylisirnio.  For small droplets of a few microns in diameter evaporation is close to the kinetic regime, in which the droplet diameter decreases linearly with time (Vlasov, 2021). This results in approximately one of magnitude difference in volatility. This is broadly consistent with Fig

8. A more thorough investigation of these effects is beyond the scope of this work and more detailed calibration work, most likely on monosized aerosol would be needed to quantify these effects.

*Line 396 "This does not impact the FIGAERO-CIMS measurements which relies on thermal decomposition.": What is meant here? FIGAERO-CIMS measurements rely on thermal desorption and thermal decomposition can and does occur in the FIGAERO, leading to multimodal thermograms which require additional processing to appropriately separate desorption and decomposition.*

Our apologies for the confusion and thanks to the referee. The FIGAERO does not rely on thermal decomposition, rather on thermal desorption. However, the referee is correct that decomposition can and does occur in the FIGAERO. The following has been added to the revised version of the manuscript *'However, Mesotrione can still be considered to be involatile as if a pesticide at the higher temperature (100°C) is non-volatile it will not be more volatile at 25°C and thus extrapolation to a lower temperature is not required'*

*Line 478-479, "Unfortunately, the reports do not specify which particular method was utilized. Consequently, it cannot be assumed that the pesticide literature values can be completely reliably compared due to the substantial variations in these methods.": Can the potential variation in methods be displayed for example in Figure 8 to capture the potential spread of literature values compared to the measurement conducted here? Are there other literature sources to display in Figure 8?*

We thank the reviewer for this point. The value is based on the regulatory framework and is not readily traceable. It is based on an upper limit value made at a higher temperature and is included for reference. We will include the sentence above to clarify this point in the revised paper. The authors recognise the importance of the suggestion of highlighting the different methods used in figure 8 in which the literature value is an upper limit made at higher temperature that is included for reference and thus shows a limitation in the results. However, a separate discussion may be suitable to understand the requirements of providing more openly available data and methods within industry, this is beyond the scope of this manuscript.

*Line 496-498, "Despite this a study in Arctic monitoring stations found low levels of Trifluralin in arctic air(Balmer et al., 2019) and has now been predicted that small amount of Trifluralin may stick to aerosol particles and transported significant distances.": This is inconsistent with your current observation of Trifluralin as an IVOC which would partition sparingly to aerosol particles. As per a comment above, please confirm that your data agrees with the Tmax and volatility characterization presented here. If it does, please discuss relevant literature and mechanisms that may allow Trifluralin to be transported to the Arctic even with a short atmospheric oxidation lifetime and high volatility.*

We thank the reviewer for their comment and understand the possible confusion of this statement. In the discussions re Trifluralin's withdrawal from use in the EU (Efsa, 2009) one of the discussions brought forward was this comment in which Trifluralin is expected to be volatile with a short oxidation lifetime and thus would not expect to be observed in the Arctic air, however despite this it has been measured at Arctic measurement stations suggesting an un-confirmed mechanism of transport is present. The authors appreciate that there may not be enough literature to support the comment highlighted by the reviewer and thus the following amendments have been made to the manuscript:

*'Studies in Arctic monitoring stations have reported low levels of Trifluralin in arctic air (Balmer et al., 2019). At present the mechanism giving rise to such long range transport are not clear, however the finding of trifluralin in such a remote environment was a major contributing factor to the removal of Trifluralin's approved status for use in the EU (Efsa, 2009).'*

**Technical Corrections**

*Line 122:* "maybe" to "may be"

This has been corrected.

*Figure 2:* Presenting these thermograms with the same horizontal axis in both panels will make the Tmax shifts more clear.

This has been corrected.

*Figure 3:* Increased marker size and consistency in marker choice for delivery method from this study across both panels would improve legibility. In 3b, I would suggest using log(vapor pressure) not ln(vapor pressure) for greater consistency across figures and more straightforward comparison to text values.

Marker choice has been made consistent. Ln is used instead to keep the axis consistent to equation 2 in which the equation of the lines represents equation 2.

*Figure 4:* Units on vertical axis.

We thank the reviewer for their comment. The vertical axis plots the dM/dlogdp from an SMPS measurement with the units of μg/cm$^3$, this has been amended.

*Figure 7:* Where error bars appear in a figure, what they represent should be listed in the figure caption. Vertical axis label should use a subscript for "max"

This has been corrected.

*Figure 8:* Increased marker size may be useful for legibility. In addition, error bars where possible should be included. Panels could be oriented side by side and use a shared legend. Finally, are multiple literature sources used in this figure? In that case, each literature source should have its own symbol and be clearly cited. If only one source is used, that should be listed in the legend and cited in the caption.

This has been corrected. Additionally, a label of a and b has been added to the figure for clarity.

*Figure 9*: This figure is extremely difficult to read, though this may be due to some upload issues. In any case, I would suggest that the labels for the points not be cut off by the frame, and use of a legend might be more appropriate anyway. I would also suggest using log(Vapor Pressure) rather than ln(vapor pressure) to make the values in the text more easily read off of the chart and for consistency with earlier figures on a log scale. Finally, labels such as "stated literature value" are not useful. Indicate which specific literature source is used for that point, I believe this would be the University of Hertfordshire Pesticide Properties Database.

We thank the reviewer for the comments and the necessary amendments have been made.

*Table 2:* Reported values should include uncertainties (standard deviation or similar).

Balmer, J. E., Morris, A. D., Hung, H., Jantunen, L., Vorkamp, K., Rigét, F., Evans, M., Houde, M., and Muir, D. C. G.: Levels and trends of current-use pesticides (CUPs) in the arctic: An updated review, 2010–2018, Emerging Contaminants, 5, 70-88, 10.1016/j.emcon.2019.02.002, 2019.

Bannan, T. J., Le Breton, M., Priestley, M., Worrall, S. D., Bacak, A., Marsden, N. A., Mehra, A., Hammes, J., Hallquist, M., Alfarra, M. R., Krieger, U. K., Reid, J. P., Jayne, J., Robinson, W., Mcfiggans, G., Coe, H., Percival, C. J., and Topping, D.: A method for extracting calibrated volatility information from the FIGAERO-HR-ToF-CIMS and its experimental application, Atmospheric Measurement Techniques, 12, 1429-1439, 10.5194/amt-12-1429-2019, 2019.

Bilde, M., Barsanti, K., Booth, M., Cappa, C. D., Donahue, N. M., Emanuelsson, E. U., Mcfiggans, G., Krieger, U. K., Marcolli, C., Topping, D., Ziemann, P., Barley, M., Clegg, S., Dennis-Smither, B., Hallquist, M., Hallquist, Å. M., Khlystov, A., Kulmala, M., Mogensen, D., Percival, C. J., Pope, F., Reid, J. P., Ribeiro Da Silva, M. A. V., Rosenoern, T., Salo, K., Soonsin, V. P., Yli-Juuti, T., Prisle, N. L., Pagels, J., Rarey, J., Zardini, A. A., and Riipinen, I.: Saturation Vapor Pressures and Transition Enthalpies of Low-Volatility Organic Molecules of Atmospheric Relevance: From Dicarboxylic Acids to Complex Mixtures, Chemical Reviews, 115, 4115-4156, 10.1021/cr5005502, 2015.

EFSA: Conclusion on pesticide peer review regarding the risk assessment of the active substance trifluralin, EFSA Journal, 7, 327r, 10.2903/j.efsa.2009.327r, 2009.

Krieger, U. K., Siegrist, F., Marcolli, C., Emanuelsson, E. U., Gøbel, F. M., Bilde, M., Marsh, A., Reid, J. P., Huisman, A. J., Riipinen, I., Hyttinen, N., Myllys, N., Kurtén, T., Bannan, T., Percival, C. J., and Topping, D.: A reference data set for validating vapor pressure measurement techniques: homologous series of polyethylene glycols, Atmospheric Measurement Techniques, 11, 49-63, 10.5194/amt-11-49-2018, 2018.

Vlasov, V. A.: On a theory of mass transfer during the evaporation of a spherical droplet, International journal of heat and mass transfer., 178, 121597, 10.1016/j.ijheatmasstransfer.2021.121597, 2021.

Ylisirniö, A., Barreira, L. M. F., Pullinen, I., Buchholz, A., Jayne, J., Krechmer, J. E., Worsnop, D. R., Virtanen, A., and Schobesberger, S.: On the calibration of FIGAERO-ToF-CIMS: importance and impact of calibrant delivery for the particle-phase calibration, Atmospheric Measurement Techniques, 14, 355-367, 10.5194/amt-14-355-2021, 2021.

---

## Author Comment (AC3)

**Responses to reviewer's #3 comments**

This study examined the volatility of pesticides using a novel approach, the Filter Inlet for Gases and AEROsols (FIGAERO) coupled with a chemical ionization mass spectrometer. Two compound delivery methods were tested, and the results were compared with those from previous studies. Volatility values were also evaluated against literature data, with potential explanations provided for discrepancies between measured and reported values. The atmospheric implications of pesticide volatilities are further discussed.

The scientific method used in this study is sound, and the results are meaningful and show promise. Additionally, the topic is significant and relevant to the field. However, I believe this manuscript is still in the draft stage and requires improvement in several key areas before it can be considered for acceptance. Overall, I would recommend rejecting the manuscript in its current form.

We'd like to thank Referee 3 for their comments and to respond to the general and detailed comments as follows (reviewer comments in black and our responses in blue; the line numbers referred throughout are referring to the original manuscript). A marked-up version of the manuscript detailing the amendments from all of the reviewers comments is also provided.

Major comments:

Many sentences in the manuscript are either grammatically incorrect or lead to confusion, making it difficult for readers to understand the content easily. For example, I've noted several specific instances, but I believe there are many others throughout the paper that require improvement. I recommend reviewing the entire manuscript to enhance clarity and readability.

Thanks to the reviewer for urging us to improve the clarity of the paper. This has been thoroughly considered as presented in the revised version.

Line 18-19: I feel it would be better if the sentence can be revised as " The pesticide volatilities were compared with widely accepted standard literature values used in industry, as well as values derived from a common environmental model frequently employed in industrial applications.

This sentence was improved for clarity and brevity according to the reviewer's comment.

Line 32-33: It may be better like this "Pesticides are a group of compounds whose fate and behaviour in the atmosphere are less studied and characterized compared to their behaviour in soil, surface water, and groundwater environments." The original sentence creates confusion by comparing the "environment" (which includes soil, surface water, and groundwater) with specific parts of the environment.

This sentence was improved for clarity by removing references to the environment. The sentence now reads the following in the amended manuscript: '*Pesticides are a group of compounds whose fate and behaviour in the atmosphere is less well studied and characterised in comparison to soil, surface water and ground water environments*'

Line 1-83: This is verbose and needs to be more concise.

We thank the reviewer for this comments. This has also been picked up by reviewer 2. We have sharpened up and reduced the verbosity of this text as presented in the marked-up version of the manuscript. We endeavour to retain the essence of the discussion in a revised version whilst sharpening the text and reducing it in size.

Line 97-106: Instead, it would be helpful to provide a more detailed introduction to the various methods used for vapor pressure measurement, as readers may be particularly interested in this aspect.

A similar comment was made by reviewer 1. A detailed response has been provided in the response to reviewer 1.

Figures: Figures are not well presented in this manuscript.

We thank the reviewer for the comment. The following has been considered in the new version of the manuscript to improve the clarity, presentation and consistency of the figures in the manuscript. Specific notes that have been improved include the consistency of brackets in the axis units of figures, the consistency of the size of the text and figures and superscript of units.

2. The workload of this study may not be sufficient for an *ACP* paper. I recommend expanding the scope by measuring more pesticides. For example, in Figure 6, we see that the Tmax for most compounds falls between 25-50°C.

We disagree strongly with this comment. The work shows how the FIGAERO-CIMS method can be applied to an important class of compounds whose vapour pressures are important in both industry and environmental science yet are very poorly defined. Previous methodologies all have some shortcomings and so a new approach is a valuable addition to the available literature. To make this more obvious the following has been added to the introduction: '*The object of this work has not been to deliver a wide-ranging study of the vapour pressures of many pesticides but rather to select a number of important pesticides based on clear criteria to demonstrate the experimental approach is a robust one and to compare with currently available literature'*. We hope that we have demonstrated the utility of the method and so future work may generate vapour pressure data for a wider set of pesticide compounds. Each of these observations is a substantial amount of work so we made sure that we selected our compounds on the following characteristics: the compounds are or have been widely used; they represent different types of pesticide classes; they have different chemical functionalities; and they also have been previously observed in the environment remote from application implying an atmospheric transport pathway. This is described clearly in section 2.4. Both other reviewers are strongly supportive of this approach. Reviewer 1 notes that *"Overall, the manuscript provides a unique measurement of pesticide volatility and is an informative first step toward building a more consistent picture of pesticide fate and transport in the environmen*t." And reviewer 2 states that "*This manuscript is of excellent scientific significance as it demonstrates the utility of the FIGAERO-ToF-CIMS measurement technique for investigating the volatility of pesticides with high environmental relevance.*".

Additionally, we would like to make the referee aware that pesticides are a group of compounds with specific functionality and thus commonly would be expected to have a Tmax within 25-50°C. The authors would also like to point the reviewer to the main theme of the manuscript which refers to atmospherically relevant compounds. The authors feel as though that choosing pesticides of a wider range of Tmax/vapour pressures would misrepresent pesticide active ingredients and go beyond the focus of this work.

There are also several alternative approaches the authors could take to make the paper more impactful and insightful:

- **Conduct additional experiments** if the current data is not ideal. For instance, with 2,4-D, if the one-order-of-magnitude difference is thought to be due to smaller particles, why not attempt to make the particle sizes more consistent with other measurements to confirm this assumption?

  Using the nebuliser and solvents that were available to us we were unable to generate a particle size distribution for 2,4D that matched the size distribution of the calibration PEG compounds. Nevertheless we felt it important to include these results in the paper rather than removing them to

show the size of the potential bias in the results if the size distributions do not agree. It further supports the discussions around aerosolization and the using appropriate drop sizes in both the calibration and observation. We have explicitly stated the shortcoming of our results for this compound and discussed potential reasons why there may be a high bias in our result. Unfortunately, we are unable to return to this work within a reasonable timeframe and so either need to remove the 2,4D results from the paper or include them with a discussion over the differences with previous literature. We have chosen the latter as we feel this adds information for the reader and future user of the work, cautioning that it is important to generate the correct size drops.

- **Dive deeper into the experimental results.** In Figure 2, why do the calibration curves for the syringe method show a more Gaussian distribution compared to the atomization method? In Figure 3b, why is linear fitting done using only a few points from Figure 3a? What are the $R^2$ values for those two fits? While I understand that the syringe method might shift Tmax due to droplet effects, does this mean the syringe method cannot provide a better linear relationship? As discussed in the paper, particle size also influences the derived vapor pressure. We really need to show more details of the experimental results before making a conclusion.

We thank the reviewer in the encouragement of some of these comments have been specifically addressed in reviewer 1 and 2's comments. However, for clarity it will be explored further here. Firstly, to support this repeated measurements we have amended the manuscript to further delve into the experimental results, this has included providing repeats of thermograms of all the experiments presented are provided in the supplementary which is accompanied by the standard deviation and full width half maximum of each peak, this aims to provide clarity on the repeatability of the experiments and the gaussian fit of the thermograms.

Additionally, the authors would like to clarify the argument presented in this paper regarding the syringe and atomisation method. Firstly, it is not that the syringe an unacceptable method. Instead, we present the atomisation method as a more suitable method to be used in this study due to the following. The atomisation method presents a more relevant method compared to online sampling in the field and thus provides an opportunity to compare against results if required. Additionally, it has been previously observed that the size of the aerosol of the calibration droplets on the FIGAERO filter need to be similar to those of the material being sampled for the calibration to be representative, in order to control this, the atomisation method is required, in which a number of small droplets are deposited on the filter, compared to one large droplet when the syringe method is used. As pointed out by Ylisirnio et al. this is due to complete evaporation of a single compound of a given volatility from the filter during the thermal ramp being dependent on the size of the particle (due to varying surface-volume ratio). A molecule in the larger droplet from the syringe method takes more energy to evaporate in comparison to the smaller droplets on the filter in the atomisation method. This results in the Tmax compound of the same volatility being higher in the syringe method and the overall thermograms being broader (as observed in figure 2 and backed up by the repeated runs provided in the supplementary), due to the spread of energies required to evaporate the compound being broader. Overall, this effect means we propose the atomisation method is more suitable. In the amendments throughout the paper, we hope the reviewer is able to further appreciate the requirement for this calibration . Consequently, due to the above arguments it must be considered that when comparing data. It is only suitable to compare the data from the same apparatus and the particles measured are the same size. In addition, this means that any calibration must have the same particle size as the compounds of interest (e.g pesticides) as the size of the compound impacts the volatility in the atomisation method.

In addition, the reason for the fitting in 3b only using a few of the points from 3a depends on the availability of literature values available (as calculated by equation 2). Specifically, This is because the Kreiger et al data only extend to vapour pressures of the PEG series up to PEG 9.

The following has been added to the text to clarify this:

*'As Kreiger et al state, this range covers all atmospherically relevant compounds that partition between gas and particle phases. As a result, while we can demonstrate that our approach to determining the Tmax of PEGs with the aerosol method can extend to larger PEGs we are unable to obtain a vapour pressure curve for these low volatilities at this stage. This analysis also demonstrates that since our thermograms closely resemble Gaussian distributions for PEGs 4 to 9 our results are representative across the whole range of relevant vapour pressures.'*

- **Extend your experiments.** How might organic mixtures affect volatility measurements? Traditionally, pure organics are used to assess volatility, but if we use mixtures at temperatures up to 200°C, will interactions between different compounds influence volatility? Additionally, is it possible to introduce these compounds into inorganic aerosol particles to examine how inorganics impact Tmax measurement?

We agree these are very interesting questions and certainly worthy of future work, but we feel, as we have expressed in our response to a previous point, that the point of this paper is clear and the evaluation is an important first step to establish the method for single compound vapour pressure measurement before mixing of particles of different volatilities can be explored more fully in future work.

3. There is no Supporting Information for this manuscript.

We included the details of the method and the results in the main body of the paper so there is no supplementary material. None of the 3 reviewers has identified detailed additional information that should explicitly be included in supplementary material. However, two reviewers have asked for further information on repeatability and statistics of the thermograms. We will show these in the revised version of the manuscript's supplementary data. In addition, this will enable comments regarding the 'goodness of gaussian fit' of the thermograms highlighted by reviewer 3 to be addressed through the representation of repeatable results.

Minor comments:

Line 1: We can consider adding Figaero in the title. Make it more specific.

We thank the referee for the opinion on the title. The authors agree with this comment and the title has been edited to read: '*Determination of the Atmospheric Volatility of Pesticides using FIGAERO - Chemical Ionisation Mass Spectrometry*'

Line 17: I'm not sure if it's appropriate to highlight "first time" here. I wouldn't claim this is the first time particle-phase pesticides have been measured with mass spectrometry. Does the article below cover particle-phase measurements? Please verify this through a thorough literature review if you still wish to use "first time." Additionally, this phrasing may cause some confusion, as it suggests pesticides were measured in field particles, whereas the study involves measuring compounds from generated particles. In other words, as long as the vapor pressure of these compounds can be measured accurately, whether the measurements are taken online or offline is not critical to this study.

Barker, Z., Venkatchalam, V., Martin, A. N., Farquar, G. R., & Frank, M. (2010). Detecting trace pesticides in real time using single particle aerosol mass spectrometry. *Analytica chimica acta*, *661*(2), 188-194.

We would like to thank the review for the comment. The line should read Particle phase Chemical Ionisation Mass spectrometry. The authors apologise for the confusion, this has been corrected for further manuscript versions.

Line 26: A lower Tmax may be corresponded to higher measured vapor pressure?

We would like to thank the reviewer for pointing out the inconsistency. The text should read '*the smaller particles deposited on the FIGAERO filter compared to the aerosolised PEG calibration particles, leading to evaporation at higher Tmax values and a lower measured vapour pressure*' This is highlighted by equation 2 and has also been previously explained by (Bannan et al., 2019) equation 1 and the difference in particle size explored by (Ylisirniö et al., 2021), which has then been theoretically explained by (Schobesberger et al., 2018). This has been amended in future versions of the manuscript.

Line 62: Please add a citation for the sentence "In terms of current EU regulatory context, …."

We thank the reviewer for this comment. The authors have added 'Regulation (EC) No. 1107/2009' as a reference. This is the legislation which details the process and requirements for a pesticide active ingredient to be allowed to be sold in the EU.

Line 78-80: We need add some citations for the sentences "there has been relatively much less attention on the fate and behaviour…", since I believe there should be some studies in this direction.

We would like to thank the reviewer for the encouragement to delve deeper into this statement. We would like to direct the reviewer to a recent review (Brüggemann et al., 2024) which highlighted the disparity between research in the soil and water environments compared to the air. Here the limitations to understanding the atmospheric portion of pesticide transport are also explored. Further to this the review points to a few studies on the fate and behaviour of pesticide in the atmosphere (Zaller et al., 2022; Butler Ellis et al., 2021; Kruse-Plaß et al., 2021). These references begin to understand different portions and mechanisms within the environment. However, these papers additionally highlight the gaps in understanding when considering atmospheric transport of pesticide. The references mentioned here have been added to phrase mentioned in the comment above.

Line 141: Where did you find this 50%-50% definition for the C*. Please cite it. I feel this is not correct.

The following has been taken from (Donahue et al., 2006) which can be found at https://doi.org/10.1021/es052297c :

'*As an example, if $C_{OA}$ =1 µg m$^{-3}$ and a given compound has $C_i^*$ = 1 µg m$^{-3}$, we would expect 50% of the mass of that compound to be found in the condensed phase and 50% in the vapor phase. In this case, any compound with (0.01≤ $C_i^*$ ≤100) µg m-3 would have a significant mass fraction in both the vapor and condensed phases*'

The reference has been added to the definition in the updated version of the manuscript.

Line 196: A polonium source should be between CH3I flow and IMR.

We thank the reviewer for the comment, this has been corrected in future versions of the manuscript.

Here I stopped looking for more minor comments and I would like to leave future work to authors.

**References**

Bannan, T. J., Le Breton, M., Priestley, M., Worrall, S. D., Bacak, A., Marsden, N. A., Mehra, A., Hammes, J., Hallquist, M., Alfarra, M. R., Krieger, U. K., Reid, J. P., Jayne, J., Robinson, W., Mcfiggans, G., Coe, H., Percival, C. J., and Topping, D.: A method for extracting calibrated volatility information from the FIGAERO-HR-ToF-CIMS and its experimental application, Atmospheric Measurement Techniques, 12, 1429-1439, 10.5194/amt-12-1429-2019, 2019.
Bilde, M., Barsanti, K., Booth, M., Cappa, C. D., Donahue, N. M., Emanuelsson, E. U., Mcfiggans, G., Krieger, U. K., Marcolli, C., Topping, D., Ziemann, P., Barley, M., Clegg, S., Dennis-Smither, B., Hallquist, M., Hallquist, Å. M., Khlystov, A., Kulmala, M., Mogensen, D., Percival, C. J., Pope, F., Reid, J. P.,

Ribeiro Da Silva, M. A. V., Rosenoern, T., Salo, K., Soonsin, V. P., Yli-Juuti, T., Prisle, N. L., Pagels, J., Rarey, J., Zardini, A. A., and Riipinen, I.: Saturation Vapor Pressures and Transition Enthalpies of Low-Volatility Organic Molecules of Atmospheric Relevance: From Dicarboxylic Acids to Complex Mixtures, Chemical Reviews, 115, 4115-4156, 10.1021/cr5005502, 2015.

Brüggemann, M., Mayer, S., Brown, D., Terry, A., Rüdiger, J., and Hoffmann, T.: Measuring pesticides in the atmosphere: current status, emerging trends and future perspectives, Environmental Sciences Europe, 36, 10.1186/s12302-024-00870-4, 2024.

Butler Ellis, M. C., Lane, A. G., O'Sullivan, C. M., and Jones, S.: Wind tunnel investigation of the ability of drift-reducing nozzles to provide mitigation measures for bystander exposure to pesticides, Biosystems engineering., 202, 152-164, 10.1016/j.biosystemseng.2020.12.008, 2021.

Donahue, N. M., Robinson, A. L., Stanier, C. O., and Pandis, S. N.: Coupled Partitioning, Dilution, and Chemical Aging of Semivolatile Organics, Environmental Science & Technology, 40, 2635-2643, 10.1021/es052297c, 2006.

Krieger, U. K., Siegrist, F., Marcolli, C., Emanuelsson, E. U., Gøbel, F. M., Bilde, M., Marsh, A., Reid, J. P., Huisman, A. J., Riipinen, I., Hyttinen, N., Myllys, N., Kurtén, T., Bannan, T., Percival, C. J., and Topping, D.: A reference data set for validating vapor pressure measurement techniques: homologous series of polyethylene glycols, Atmospheric Measurement Techniques, 11, 49-63, 10.5194/amt-11-49-2018, 2018.

Kruse-Plaß, M., Hofmann, F., Wosniok, W., Schlechtriemen, U., and Kohlschütter, N.: Pesticides and pesticide-related products in ambient air in Germany, Environmental Sciences Europe, 33, 10.1186/s12302-021-00553-4, 2021.

Schobesberger, S., D'Ambro, E. L., Lopez-Hilfiker, F. D., Mohr, C., and Thornton, J. A.: A model framework to retrieve thermodynamic and kinetic properties of organic aerosol from composition-resolved thermal desorption measurements, Atmospheric Chemistry and Physics, 18, 14757-14785, 10.5194/acp-18-14757-2018, 2018.

Ylisirniö, A., Barreira, L. M. F., Pullinen, I., Buchholz, A., Jayne, J., Krechmer, J. E., Worsnop, D. R., Virtanen, A., and Schobesberger, S.: On the calibration of FIGAERO-ToF-CIMS: importance and impact of calibrant delivery for the particle-phase calibration, Atmospheric Measurement Techniques, 14, 355-367, 10.5194/amt-14-355-2021, 2021.

Zaller, J. G., Kruse-Plaß, M., Schlechtriemen, U., Gruber, E., Peer, M., Nadeem, I., Formayer, H., Hutter, H.-P., and Landler, L.: Pesticides in ambient air, influenced by surrounding land use and weather, pose a potential threat to biodiversity and humans, The science of the total environment., 838, 156012, 10.1016/j.scitotenv.2022.156012, 2022.